# Engineering synthetic cells with intramembrane domains possessing distinct bilayer asymmetries

Naresh Yandrapalli [1,2] ✉, Tina Seemann[1], Reinhard Lipowsky [1] & Tom Robinson [1,3] ✉

Our understanding of how membrane asymmetry governs biological function is limited by the lack of techniques to produce model membranes which can reliably and accurately mimic cellular membrane asymmetry. Not only in terms of asymmetric lipid distribution, but also how that asymmetry can be confined to specific lateral locations across the membrane. Here we present an inverted emulsion method that can be used to produce synthetic cells with symmetric and asymmetric bilayers, as well as phase separation where the intermembrane domains possess distinct bilayer asymmetries. We assess the degree of lipid asymmetry using protein-lipid interaction and quenching assays. Surprisingly, the synthetic cells with asymmetric and phase separated membranes displayed pronounced curvature of the domains and resulted in membrane budding and division. Overall, this work develops biomimetic membranes with lipid compositions akin to natural biomembranes – an essential element in the development of functional synthetic cells.

The asymmetric nature of biomembranes is observed for all membranes present in eukaryotic cells[1–4]. With distinct lipid compositions in its leaflets, the plasma membrane is selective with specific functionalities[5,6]. For example, its inner leaflet is negatively charged, caused by a majority of charged lipids like phosphatidylserine (PS) and phosphatidylinositol phosphates. The presence of these lipids in the inner leaflet allow important cellular activities like signaling cascades, membrane protein interactions and assemblies such as ESCRTs and even viral proteins[7,8]. In contrast, the outer leaflet is devoid of such negatively charged lipids and instead containing mostly sphingolipids and glycolipids, which play a significant role in the formation of lipids rafts/domains and in cell-to-cell communication[9,10]. In fact, it is understood that having charged lipids like PS in the outer leaflet of the plasma membrane is a sign of cellular apoptosis[11–13]. While PS is indeed absent from the outer leaflet of healthy cell membranes, the outer leaflet can still possess a net negative charge due to the presence of alternative negatively charged components, such as gangliosides or sialylated glycolipids. These lipids contribute to the overall negative surface potential of the outer membrane, which is critical for biological processes such as cell-cell interactions and the efficacy of drug delivery systems utilizing positively charged polymers or lipids[14]. Recently, molecular dynamic simulations of a fully asymmetric plasma membrane were performed, excluding protein asymmetry, to understand lipid organization[15]. While computer simulations can reveal interesting insights into the lipid organization and their diffusion, it is still necessary to produce experimental asymmetric model membranes (including the associated membrane proteins) to not only understand various biophysical phenomena (e.g., membrane re-modeling) but also to mimic features of eukaryotic cells in the context of bottom-up synthetic biology.

Despite our knowledge of the complex and extensive lipid composition of the cellular membrane, as obtained from lipidomic studies, building asymmetric model membranes in the lab is still challenging[16]. Methods such as droplet interface bilayers (DIBs)[17], vesicle (hemi) fusion to a supported lipid bilayer (SLB)[18], or the use of methyl-β-cyclodextrin[19] offer ways to produce asymmetric bilayers but the

[1]Theory & Bio-Systems Department, Max Planck Institute of Colloids and Interfaces, Potsdam, Germany. [2]Faculty of Natural Sciences and Technology, University of Saarland, Saarbrucken, Germany. [3]Institute for Bioengineering, School of Engineering, University of Edinburgh, Edinburgh, UK. ✉e-mail: naresh.yandrapalli@uni-saarland.de; tom.robinson@ed.ac.uk

resulting membrane structures are limited. In the case of DIBs, the presence of large amounts of oil can be an issue for specific proteins or lipids, and subsequent manipulation (e.g., to add biomolecules from outside) after the formation of the bilayers is not possible. In the case of vesicle fusion-based asymmetry generation, usage of divalent cations/oppositely charged lipids could add to the complexity. Moreover, the resulting SLB or giant unilamellar vesicle (GUV) has an inaccessible inner leaflet or requires lipid flip-flop to alter its composition. The methyl-β-cyclodextrin method can produce asymmetric membranes but is limited to specific lipid types with limited access to the vesicle lumen and is process intensive. For these reasons, cell-sized free-standing GUVs with accessible outer and inner leaflets are desired. The latter point is an important one as not all GUV production methods provide high encapsulation. Being able to visualize them and encapsulate larger biomolecules within makes them a strong candidate for building synthetic cells from the bottom-up[20]. Techniques like electroformation and hydration methods only yield symmetric bilayers and are further limited by difficulties with using physiological buffers or low encapsulation efficiencies. Encapsulation is essential when modeling cells, therefore, researchers require not only asymmetric membranes but also high encapsulation of various biomolecules[21]. More sophisticated techniques such as microfluidics have been shown to create GUVs with a variety of lipid asymmetries with excellent encapsulation efficiencies[22–25]. For example, pulsed-jetting technique was used to create cell-size lipid vesicles with asymmetric lipid bilayers[26], and double emulsion-based templating is adopted to create lipid vesicles with biotinylated lipid asymmetry[27] as well as bacteria-like vesicles with lipopolysaccharides in the outer leaflet[28]. Microfluidic platforms have the advantage of precision in size, encapsulation efficiency, along with a high degree of asymmetry. However, the experimental complexities of microfluidic setups have curtailed their widespread applicability, leading to the requirement of a simple yet effective method to assemble asymmetric model membranes.

Previously, our lab and others have shown that the inverted emulsion method can be optimized to efficiently produce GUVs that can also encapsulate large biomolecules, while still being accessible and time effective (<20 min)[29–33]. Using this technique, researchers have developed thermoresponsive symmetric vesicles and light activated nanopore assemblies for cargo release applications[34,35]. Functional applications in the form of artificial cell-based therapeutic protein delivery for tumor treatment and tissue angiogenesis was achieved[36,37]. Furthermore, complex multi-layered membrane systems via layer-by-layer assembly was proposed through the inverted emulsion method[38]. Despite the recent applications of the inverted emulsion method, the adoption of this technique remains limited amongst the biophysical community for asymmetric membrane applications. To engineer realistic synthetic cells with asymmetric membranes akin to the plasma membrane, it is necessary to improve the technique for reproducible production of vesicles with various lipid types, functional proteins, and even to include lipid domains.

Previous research in this direction has shown that it is possible to produce phase-separated GUVs using the inverted emulsion method but with limited success in cholesterol incorporation or reproducibility[35,39,40]. Presently, to our knowledge, there is no methodology that has been shown to produce asymmetric membranes, along with the complexity of phase separation that can also sustain protein functionality. Here, we address this shortfall and present a methodology to achieve asymmetric membranes with lipid domains and functional protein interactions.

To achieve this, we developed a triple-layer inverted emulsion method with a layer-by-layer addition of the solutions which allows production of asymmetric and/or phase-separated GUVs. First, the outer aqueous solution is layered with an oil solution containing an outer leaflet lipid composition. This is followed by layering of a 'spacer' oil (without lipids) and finally by a pre-formed water-in-oil (W/O)

emulsion containing the inner leaflet lipid composition in the oil phase. After a quick centrifugation step, asymmetric GUVs are produced when the W/O droplets with the inner leaflet lipids pass through the interfacial monolayer containing the outer leaflet lipids. We measured the degree of asymmetry of the formed lipid bilayers via a dye quenching assay and further confirmed it through protein-lipid binding studies. Various proteins (streptavidin/GFP/Cholera toxin subunit B) either alone or in combinations were tested to asymmetrically decorate (inside and outside) GUVs tagged with specific ligand-lipid types (Biotinyl Cap PE/DGS-Ni-NTA/GM1). To demonstrate the flexibility of the method, we show the production of more complex asymmetric lipid membranes exhibiting not only phase-separation but also asymmetry in peripheral proteins. Upon formation of such complex membranes, we report synthetic cells with out-of-equilibrium morphologies leading to membrane budding and division (under osmotic deflation). By theoretical considerations, our results suggest that asymmetry in the bilayer in combination with domains can exhibit (two different values of) spontaneous curvature contributing to the formation of a closed membrane neck that undergoes fission.

## Results

### Triple-layer inverted emulsion method

To achieve a high degree of bilayer asymmetry in GUVs, the challenge is to prevent mixing of the two different lipid containing oil solutions. Failure to do so will result in contamination of the inner and outer leaflets with the opposite lipids and decrease the degree of asymmetry. For this reason, we have carefully selected and employed two different oils with varying densities (to avoid mixing of oils during the centrifugation step), and crucially, we use an empty oil as a spacer layer (free from lipids) in between them (Fig. 1). This avoids any lipid mixing during the production process.

After the preparation of the lipid oils containing two distinct lipid compositions for inner and outer leaflets (Fig. 1), the outer leaflet lipid composition containing squalene lipid oil is added to the microtiter plate well with outer aqueous solution (bottom left box, Fig. 1). The entire set is allowed to incubate for 30 min to form the interfacial monolayer as the lipids self-assemble along the squalene-water interface. Following this, lipid-free mineral oil is gently spread over the squalene layer, which acts as a spacer oil as described earlier. Immediately, a W/O emulsion formed from the second lipid composition containing mineral oil and the inner solution (sucrose solution with or without streptavidin, top box Fig. 1) is spread on top of the spacer oil (right, Fig. 1). Finally, centrifugation is performed to assist the movement of the inner leaflet lipid composition stabilized W/O droplets in the top-most layer towards (and through) the interface monolayer to produce vesicles with asymmetric bilayers. The same technique is used to produce symmetric bilayers by using the same lipid composition in both oil phases.

Typical examples of the resulting symmetric and asymmetric GUVs can be seen in Fig. 2a, b, respectively. Despite the change in the lipid composition for symmetric and asymmetric GUVs, there is no substantial difference in the morphology or the yield of the GUVs produced. In this method, an incubation step of 30 min is employed for the formation of interfacial monolayer to produce asymmetric GUVs. Contrary to previous methods for symmetric GUVs, we found that it is important to include an incubation step as well as an extra spacer oil to prevent any possible adulteration of the bilayers (through mixing of the oils and lipids)[41]. In the present work, as the W/O droplets pass through the oil-water interface (i.e., squalene-glucose), the interfacial monolayer will be depleted of lipids. Any further W/O droplets crossing the interface, in places without the interfacial monolayer, can still result in vesicles but they will only have excess oil and contain membranes that are symmetric or with low asymmetry. To avoid this, either the surface area of the interface between squalene and glucose solution has to be increased or the number of W/O

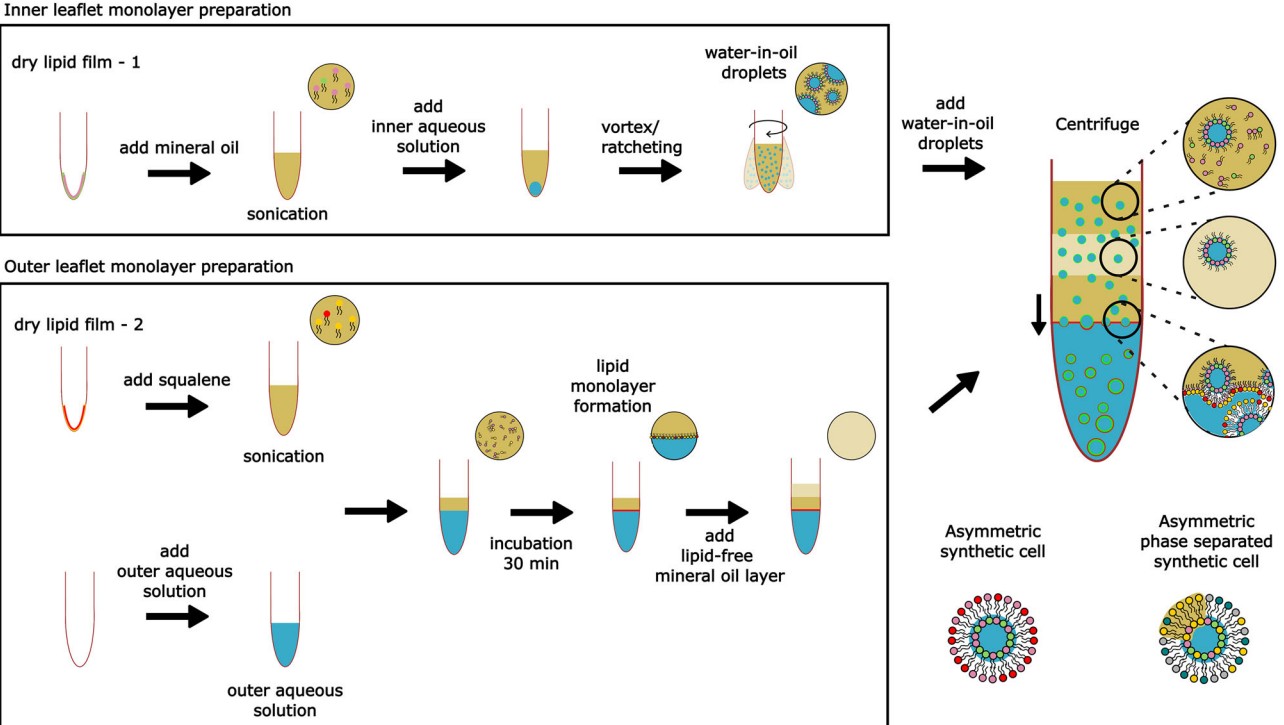

**Fig. 1 | Schematic illustration of the triple-layer inverted emulsion method to produce asymmetric (or symmetric) GUVs with (or without) phase separation.** First, preparation of two sets of lipid oils with distinct lipid compositions via sonication. Followed by the addition of lipid oil with outer leaflet composition to the vial with the outer solution and incubating it for 30 min to form the interface monolayer before adding the empty spacer mineral oil (no lipids). Later, addition of lipid oil with inner leaflet composition to an Eppendorf® tube containing the encapsulate or inner aq. solution for forming the W/O emulsion is conducted. Finally, pre-formed W/O emulsions are added to the vial with interfacial monolayer and centrifuged to yield GUVs (symmetric or asymmetric or asymmetric phase separated).

droplets needs to be reduced. However, increasing the concentration of the lipids does improve monolayer formation in general, but beyond 200 μM we do not see a significant improvement[33]. Considering, the time implemented to form the lipid monolayer is 30 min and the centrifugation time is 3 min, the rate of formation of lipid monolayer is much longer than the rate of droplet transfer. For this reason, we reduced the number of W/O droplets by using only 3 μL of the inner aqueous solution (compared to 5 μL from our previous work[33]).

The asymmetry can also be compromised by contamination of inner monolayer lipids with outer ones and vice versa. If too many excess lipids from the emulsion remain, they can also get pulled down with the W/O droplets and end up in the outer leaflets of the GUVs. Here, the presence of a oil spacer (i.e., mineral oil without lipids) dilutes the lipids from W/O droplet in excess mineral oil before passing into the squalene layer and ensures a high degree of asymmetry in the final bilayers (Fig. 2c). Our previous work, without these precautionary steps to prevent contamination, has shown a lower degree of asymmetry when compared to the results of this work[32].

**Fluorescence quenching assay to assess bilayer asymmetry**

Fluorescence quenching is an established method to assess the degree of asymmetry in lipid bilayers. In our experiments, we performed a quenching assay using NBD-tagged lipids and sodium dithionate as a quencher - a commonly used combination[42]. The quenching occurs when sodium dithionite, a reducing agent, interacts with the fluorescent NBD (7-nitrobenz-2-oxa-1,3-diazol-4-yl) group, reducing it to a non-fluorescent form. This reaction is highly selective and depends on the accessibility of the NBD fluorophore to the aqueous dithionite, providing a measure of its exposure in membrane systems[43]. In the first set of experiments, we compared the quenching percentage in symmetric GUVs produced from the previously reported double-layer

method to our triple-layer method. The lipid composition for this study was 99.5 mol% POPC and 0.5 mol% NBD-DOPE. As displayed in Fig. 2c, the fluorescence intensity of the symmetric GUVs produced through the double-layer method is reduced to $62 \pm 13\%$ from the unquenched membrane and a similar percentage of $53 \pm 11\%$ is observed for the GUVs produced using the triple-layer method, after the addition of sodium dithionate. This result is anticipated as the NBD-DOPE lipid gets quenched on the outer leaflet of the symmetric bilayer only, and the intensity should reduce to half. From this data, statistically, it can be concluded that no significant difference in the percentage of quenching is observed amongst the symmetric GUVs produced from both the methods and quenching assay can be reliably performed on the asymmetric GUVs.

Two different populations of asymmetric GUVs were then produced using both methods. One comprised of GUVs produced with NBD-DOPE in the inner leaflet of the GUVs and the other had NBD-DOPE in the outer leaflet. Like the previous condition, sodium dithionate was added to the outer solution for both populations of asymmetric GUVs. In the case of NBD-DOPE in the inner leaflet, no or very few of the GUVs should be quenched for successful asymmetricity. However, for the case when NBD-DOPE is in the outer leaflet (and therefore accessible to the quencher), all the fluorescence intensity should be quenched. A time-lapse video showing the quenching of asymmetric GUVs (with NBD-DOPE in the outer leaflet) in the presence of sodium dithionate can be seen in the supplementary information (Supplementary Video 1). In Fig. 2c, for the asymmetric NBD inside condition, both methods exhibit low quenching: double-layer $(14 \pm 11\%)$ and triple-layer $(7 \pm 3\%)$. While in the asymmetric NBD outside condition, quenching is high for both methods with $77 \pm 6\%$ for double layer method and $97 \pm 1.3\%$ for triple layer method. Interestingly, the triple-layer methodology shows lower variability in the degree of asymmetry. These results in conjunction with statistical

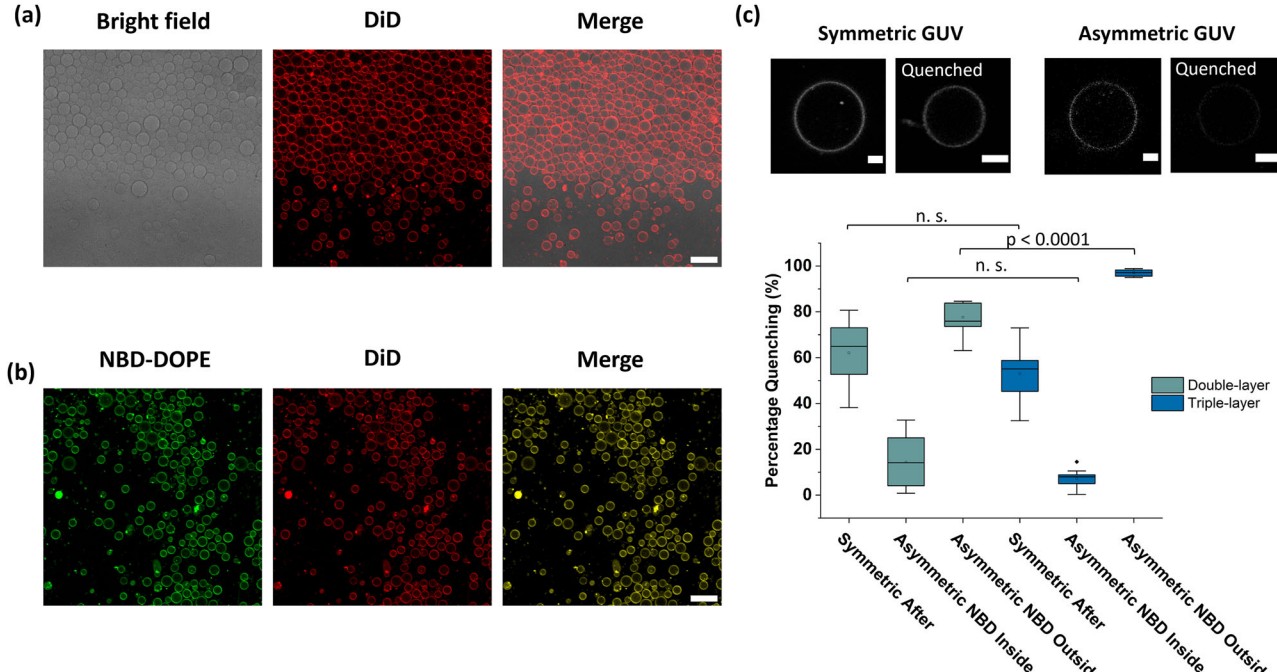

**Fig. 2 | Asymmetric (and symmetric) GUVs produced using the triple-layer inverted emulsion method. a** Confocal images of symmetric GUVs (99.5 mol% POPC and 0.5 mol% DID (red color)) and **b** asymmetric GUVs (99 mol% POPC in both of the leaflets of the bilayer with 0.5 mol% NBD-DOPE (green color) in the inner leaflet and 0.5 mol% DID (red color) in the outer leaflet). Scale bars correspond to 50 μm. **c** The change in fluorescence intensity for symmetric GUVs and asymmetric GUVs induced by quenching from the addition of 10 mM sodium dithionate to the outer solution. Inserts show confocal images of exemplary GUVs before and after quenching produced via the triple-layer method ($n \leq 30$, per every condition, three or more independent experiments are performed). Scale bars correspond to 5 μm. Statistical significance was determined using two-sided Mann–Whitney U test. Box plots: center line = median; box = Q1 (25th percentile)–Q3 (75th percentile); whiskers = lowest and highest values within 1.5×IQR of Q1/Q3 (Tukey); points beyond whiskers = outliers; filled dot = mean. Min/Max denote the smallest/largest observations. Error bars indicate standard deviation (SD). Data is expressed as the mean±SD. Source data are provided as a Source Data file.

analysis suggest that the two production methods lead to distinct quenching profiles under the tested conditions, with the triple-layer method demonstrating more consistent results (less variability) and improved asymmetry compared to the double-layer method. To evaluate the hypothesis that the presence of spacer oil prevents lipid mixing and improves the asymmetry, we added labels to each layer. Observations are made of the mixing of the labels after the centrifugation step. The Supplementary Fig. 1 shows that the double-layer method results in the homogeneous mixing of upper mineral oil layer with the squalene layer, and a uniform color. In the case where the triple-layer method, but without spacer, some mixing is observed. Only in the case of the triple-layer with spacer, no oil mixing was observed. This confirms that a reduced lipid mixing in the methodology contributed to the improved asymmetry, ~95% compared to ~80% or lower that is typically observed amongst works using the double-layer method[32,43,44]. In addition to the density differences which maintain the separation of mineral oil and squalene, this test emphasizes the role of spacer oil in further reducing the mixing of lipids, ensuring high lipid asymmetry across the membrane.

**Protein-lipid bilayer asymmetry**

Protein-lipid interactions are universal and play an important role in cellular activities. To demonstrate the functionality of the produced asymmetric synthetic cells, we utilized specific proteins with a high affinity towards ligand-tagged lipids that are present only in one bilayer leaflet. First, the high affinity of streptavidin towards biotin was exploited to not only utilize the asymmetry of the membranes, but also to demonstrate the assembly of a biomimetic GUV using this technique. In Fig. 3a, streptavidin (green channel) is encapsulated within the lumen of the vesicles while biotin-tagged lipids are only present in the

outer leaflet of the membrane. Note that this configuration is only made possible because the inverted emulsion method can easily encapsulate large biomolecules within the GUVs along with the advantage of producing asymmetric lipid membranes. Importantly, the streptavidin protein does not bind to the GUVs as the biotin-tagged lipids are only in the outer leaflet (see also Supplementary Video 2). Indeed, the line segment analysis proves the clear demarcation between the green and red signals of the protein and lipid bilayer (Supplementary Fig. 2a). However, it is possible that the vesicle bilayers were contaminated with oil which could potentially prevent the binding of streptavidin. To investigate this possibility, streptavidin was added externally to the same set of GUVs (see Fig. 3b and Supplementary Video 3). Streptavidin can indeed bind to the biotin-tagged lipids present in the outer leaflet based on the intense green fluorescence observed along the lipid bilayer resulting from protein enrichment. This is further confirmed by line segment analysis revealing clear overlap of the green and red regions indicating binding (Supplementary Fig. 2b). This observation not only confirms that the membranes produced using the triple-layer method are highly asymmetric, but also biomimetic in nature (i.e., lipid bilayers without oil contamination) and can be functionalized.

Further experiments were performed with a different protein-ligand-tagged lipid system to establish the universal applicability of the technique. It is well understood that streptavidin-biotin interaction is one of the strongest non-covalent interactions in nature, while Ni-NTA-histidine tag interactions typically have a disassociation constant in μM range. Thus, comparing weaker and reversible coordination interactions of Ni-NTA presents a compelling proof of the biomimetic and functional asymmetry of the bilayer vesicles. Therefore, we employed His-tagged GFP and Ni-NTA-tagged lipids. The lack of signal at the

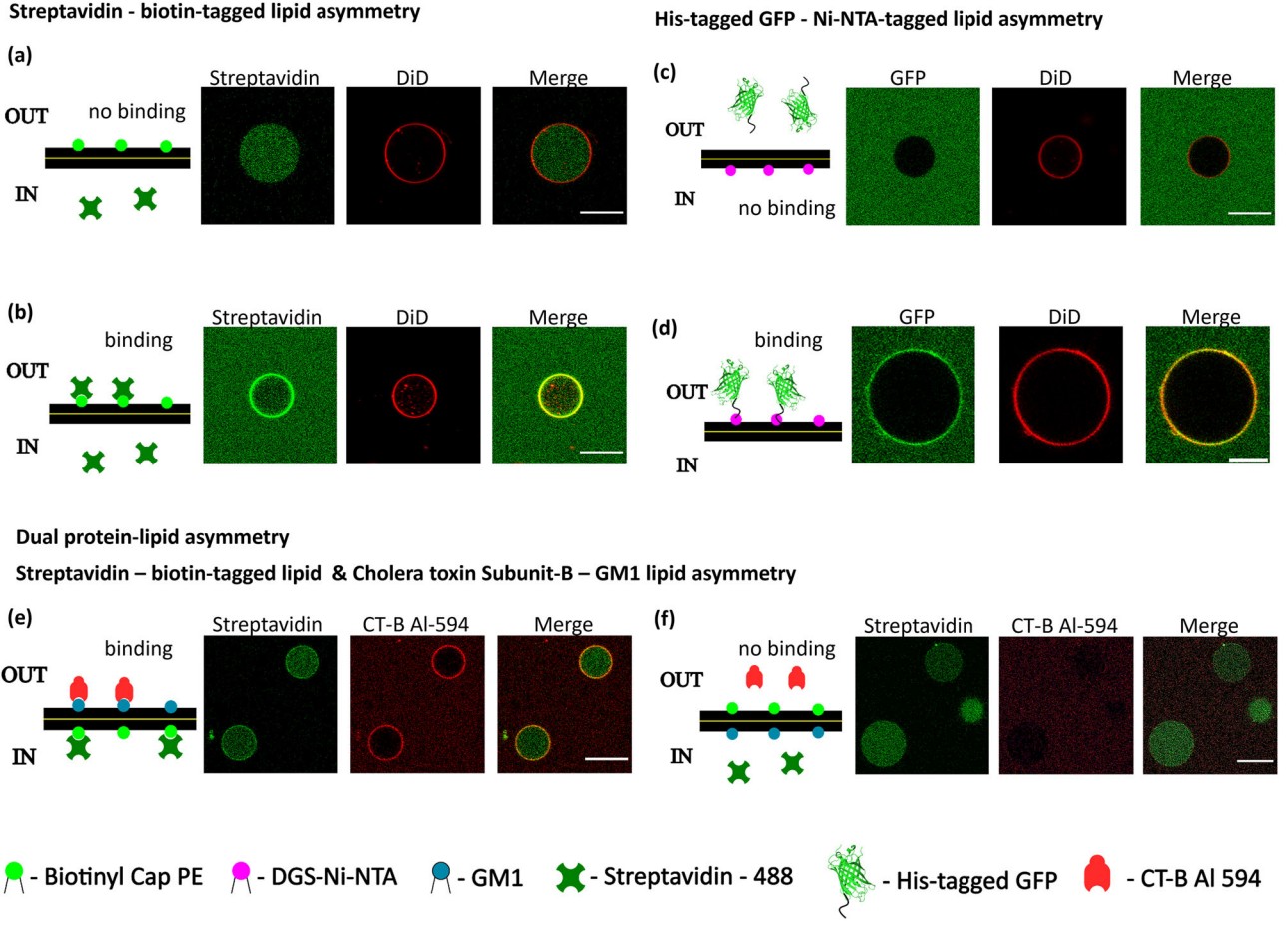

**Fig. 3 | Functional membrane asymmetry demonstrated with protein-lipid interactions. a** Confocal images of asymmetric GUVs (99.5 mol% POPC and 0.5 mol% DiD (red color) in the inner leaflet and 99.5 mol% POPC and 0.5 mol% Biotinyl Cap PE in the outer leaflet) with unbound alexa 488-tagged streptavidin (green color) in the lumen of the GUVs because the biotinylated lipids are present only in the outer leaflet, negative control. **b** Confocal images of the same GUVs as presented in (**a**) after the addition of alexa 488-tagged streptavidin (green color) in the outer solution. **c** Confocal images of His-tagged GFP in the outer solution of vesicles with 99.5 mol% POPC and 0.5 mol% DiD (red color) in the outer leaflet and 0.5 mol% DGS Ni-NTA in the inner leaflet, negative control. **d** Confocal images of His-tagged GFP (green color) in the outer solution of vesicles with 99.5 mol% POPC and 0.5 mol% DiD (red color) in the inner leaflet and 0.5 mol% DGS Ni-NTA in the outer leaflet.

**e** Confocal images of GUVs with streptavidin (green color) (inside) and cholera toxin subunit-B tagged with AlexaFluor® 594 (CT-B Al-594) (red color) (outside) binding to biotinylated lipids present in the inner leaflet and to GM1 present in the outer leaflet of GUVs respectively (99.5 mol% POPC and 0.2 mol% biotinyl Cap PE in the inner leaflet and 1 mol% GM1 in the outer leaflet). **f** Confocal images of GUVs (negative control) with streptavidin (green color) (inside) and CT-B Al-594 (red color) (outside) not binding to GM1 present in the inner leaflet and biotinylated lipids present in the outer leaflet of GUVs respectively (99.5 mol% POPC and 0.2 mol% biotinyl Cap PE in the outer leaflet and 1 mol% GM1 in the inner leaflet). Corresponding schematics are shown to the left of each image. $n \geq 30$, data is collected from three or more independent experiments. Scale bars correspond to 20 μm.

membrane shown in Fig. 3c demonstrates that the GUVs with Ni-NTA lipids in the inner leaflet are successfully separated from the His-tagged GFP present in the outer solution due to the asymmetric nature of the vesicles (see also Supplementary Video 4). On the other hand, binding is observed when both Ni-NTA-tagged lipids and His-tagged GFP are located on the same side (here, outside) of the membrane, which can be seen in Fig. 3d (see also Supplementary Video 5). The line segment analysis clearly indicates the localization of the protein along the lipid bilayer when DGS Ni-NTA lipid is presented on the outer leaflet of the GUVs (Supplementary Fig. 2d) and no co-localization when the lipid is in the inner leaflet (Supplementary Fig. 2c). These results not only verify the asymmetric nature of the resulting membranes but also demonstrate that it is possible to produce functional asymmetric synthetic cells using different ligand-tagged lipid−protein pairs.

To further demonstrate the flexibility of the method, a more complex configuration of asymmetric GUVs is produced with two different ligand-tagged lipids residing in the opposite bilayer leaflets, and as above, both binding and non-binding experiments are

performed. First, vesicles are prepared with biotin-tagged lipids in the inner leaflet and GM1 in the outer leaflet (see Fig. 3e). During the production process, and taking advantage of the inverted emulsion method, streptavidin (green channel) is encapsulated within the vesicles while cholera toxin subunit-B (red channel) is added to the external solution. Consequently, both proteins are observed to bind successfully to their respective ligand-tagged lipids, i.e., streptavidin binding to the biotinyl cap PE in inner leaflet and CT-B AL-594 to the GM1 in outer leaflet of the vesicles. A negative control is also performed whereby the leaflet positions of the ligand-tagged lipids are reversed while keeping both the proteins in their respective configurations, i.e., streptavidin inside the vesicles and CT-B AL-594 outside while biotin-tagged lipids are in the outer leaflet (away from streptavidin) and GM1 in the inner leaflet (away from CT-B AL-594). The data show that neither of the proteins bound to the vesicle membrane (Fig. 3f). Again, the line segment analysis further proves our observation that the produced GUVs are indeed asymmetric and allow for specific interactions in the presence of the two proteins (Supplementary Fig. 2e, f).

Finally, we quantified the results for all three distinct molecular recognition systems in Fig. 3. For each system, the positive control was when the protein in the external solution could access its target lipid on the outer leaflet, and the negative control was when the target lipid was situated on the inaccessible inner leaflet (Supplementary Fig. 3). The results demonstrate a high degree of binding specificity. For the streptavidin-biotin system, we observed binding in 89.5% of the positive control GUVs, whereas only 6.6% of negative control vesicles showed a signal. The specificity was even more pronounced for the His-tag GFP-NTA system, which resulted in 100% binding for the positive control and 0% for the negative control. Finally, we tested the binding of the CT-B to its target ganglioside, GM1. A robust binding signal was detected in 97.0% of GUVs when GM1 was on the outer leaflet, compared to only 9.1% when it was on the inner leaflet. Not only does this demonstrate the method's flexibility to produce asymmetric GUVs with two different leaflet specific lipids able to bind different proteins, but this experiment showcases its ability to combine two asymmetries: both within the membrane (i.e., inner and outer leaflets), and across the membrane (i.e., inner and outer solutions).

## Phase separation using sphingomyelin

Lipid partitioning in membranes is the major contributor to raft formation in cell membranes. In association with various proteins that can influence the clustering of lipids, transbilayer coupling has been shown to influence domain formation in membranes[9,45]. In this context, synthetic cells also require complex lipid mixtures to produce phase-separated membranes (i.e., liquid-ordered (Lo) and liquid-disordered (Ld) domains) as well as being able to encapsulate biomolecules in the lumen to study transmembrane signaling pathways, transbilayer molecular interactions and protein assemblies. While electroformation based production of phase separated GUVs is a common strategy, it is challenging to produce them using inverted emulsion method. Previous studies demonstrated very poor incorporation of cholesterol when mineral oil was used[39,46] except for one study[35] (see Supplementary Table 4). However, we are unable to reproduce the phase separation using double layer method involving mineral oil alone (Supplementary Fig. 4). This inconsistency motivated us to develop an alternative method. Using the triple-layer method, we are successful in producing phase-separated GUVs from a lipid mixture containing POPC, cholesterol (chol) and sphingomyelin (SPM) (see Fig. 4).

The usage of squalene, instead of the commonly used mineral oil, paved the way for successful incorporation of cholesterol into the lipid membranes. This is because the solubility of lipids (saturated lipids such as sphingomyelin) and cholesterol in mineral oil is high, thanks to the presence of long-chain saturated hydrocarbons. The extent of phase separation and confocal images of the phase-separated GUVs are presented in Fig. 4b, c. GUVs were formed with varying concentrations of cholesterol (in both the squalene and mineral oil phases) while keeping the sphingomyelin to DOPC ratio constant (i.e., 1:1). The data is presented as a phase diagram in Fig. 4a and confocal images of the GUVs with varying percentages of cholesterol content can be seen in Fig. 4c. Beyond 60% of cholesterol, we observed no further phase separation (i.e., the majority had no visible domains). The membranes are predominantly with single-phase for GUVs containing 70%, 80% and 90% cholesterol content (Fig. 4a) as observed elsewhere[35].

Additionally, we compared these results to those made by the most used method to produce phase-separated GUVs, the electroformation method (Fig. 4b). Unlike similar studies, our method provides improved agreement with the results obtained by the electroformation method. Due to high solubility of cholesterol in mineral oil, studies using mineral oil have shown poor cholesterol membrane incorporation resulting in predominantly gel-phased lipid domains[35,46]. This signifies the importance of modifying the inverted emulsion method to replicate the gold standard, i.e., electroformation. The triple-layer method presented in this study achieves this while still

being able to produce asymmetric GUVs, both membrane and solutions. Note that the electroformation method cannot produce asymmetric GUVs (neither bilayer nor solution asymmetry). Within the tested concentrations, we are also able to produce phase-separated GUVs with liquid–liquid co-existing phases by varying sphingomyelin concentration (while fixing the DOPC:Chol ratio at 1:1) (Supplementary Fig. 5). Taken together, these results provide strong evidence for the formation of phase-separated vesicles using the triple-layer method. Unlike previous work using the inverted emulsion method, the study here is exclusively focused on sphingomyelin as a major contributor for the domain partitioning and not DPPC[35,39]. Sphingomyelin is one of the major biomembrane lipids and exists in high concentrations in the outer leaflet of the plasma membrane. Moreover, along with cholesterol and various proteins, sphingomyelin has been proposed to play a significant role in the formation of rafts or lipid domains in the plasma membrane[9]. Our results present a step forward in the direction of modeling and studying rafts/domain creation via protein assemblies and transbilayer interactions.

## Production of asymmetric phase-separated membranes

The triple-layer method also allows combining the above to produce complex asymmetric phase-separated vesicles. The schematic shown in Fig. 5a depicts the trans-bilayer asymmetry of lipids confined to the Ld and Lo domains of the GUV. In the Ld domain, asymmetry is achieved through selective addition of biotinyl cap PE lipid to the inner leaflet of the Ld domain only. In the case of Lo domain, GM1 lipid is confined to the outer leaflet of the Lo domain only. This is confirmed through the binding of streptavidin and CT-B Al-594 proteins that have high specificity towards biotin and ganglioside (Fig. 5a and Supplementary Fig. 6). However, the addition of both streptavidin (inside) and CT-B Al-594 (outside) proteins further add to the asymmetry of the system. Such a design mimics the natural asymmetry present within biological membranes (distinct lipid types and peripheral proteins on either side of the membrane), creates distinct internal and external functionalities, and broadens the system's use for modeling of biomembranes. For example, this setup could be used to study the selective binding of signaling proteins to one side of the membrane while simultaneously investigating receptor-ligand interactions or membrane curvature on the external membrane surface. In general, we believe that more complex asymmetric membranes will be crucial for increasing the functionality of synthetic cells, either within the field of bottom-up synthetic biology or for applications in biotechnology and biomedicine[47]. Indeed, upon addition of CT-B Al-594 two distinct spontaneous curvatures are observed, here. Figure 5b shows confocal images of asymmetric phase-separated GUVs incubated with CTB subunit (tagged with Alexa-594 dye, red). The asymmetric phase-separated GUVs have lipid composition that induces phase-separation (DOPC:SPM:Chol - 1:1:2, with 0.2 mol% biotinyl cap PE in the inner leaflet, and 1 mol% GM1 in the outer leaflet). The lipid asymmetry is achieved through doping of the inner leaflet with 0.2 mol% biotinyl cap PE which partitions into the Ld phase and the outer leaflet with 1 mol% of GM1 which partitions into the Lo phase[48]. This domain-specific asymmetry resulted in unexpected membrane dynamics within the asymmetric phase-separated GUVs that have not been reported in literature. Unlike the vesicles with symmetric bilayers, the asymmetric phase-separated GUVs exhibited a noticeable outward protrusion after the binding of CT-B Al-594 to the GM1 within the Lo domain (Fig. 5b, c and Supplementary Video 6). Typically, Lo domains are tightly packed and are less flexible compared to Ld domains[49,50].

Here, the Lo domain protrudes toward the GM1-containing outer leaflet after the addition of CT-B Al-594 (Fig. 5b, c, and Supplementary Video 6). The outward-budding of the smaller Lo domain acts to lower the line free energy of the domain boundary between the two domains[51,52]. In contrast, when adding CT-B Al-594 to symmetric phase-separated GUVs, no such protrusion was observed (Supplementary

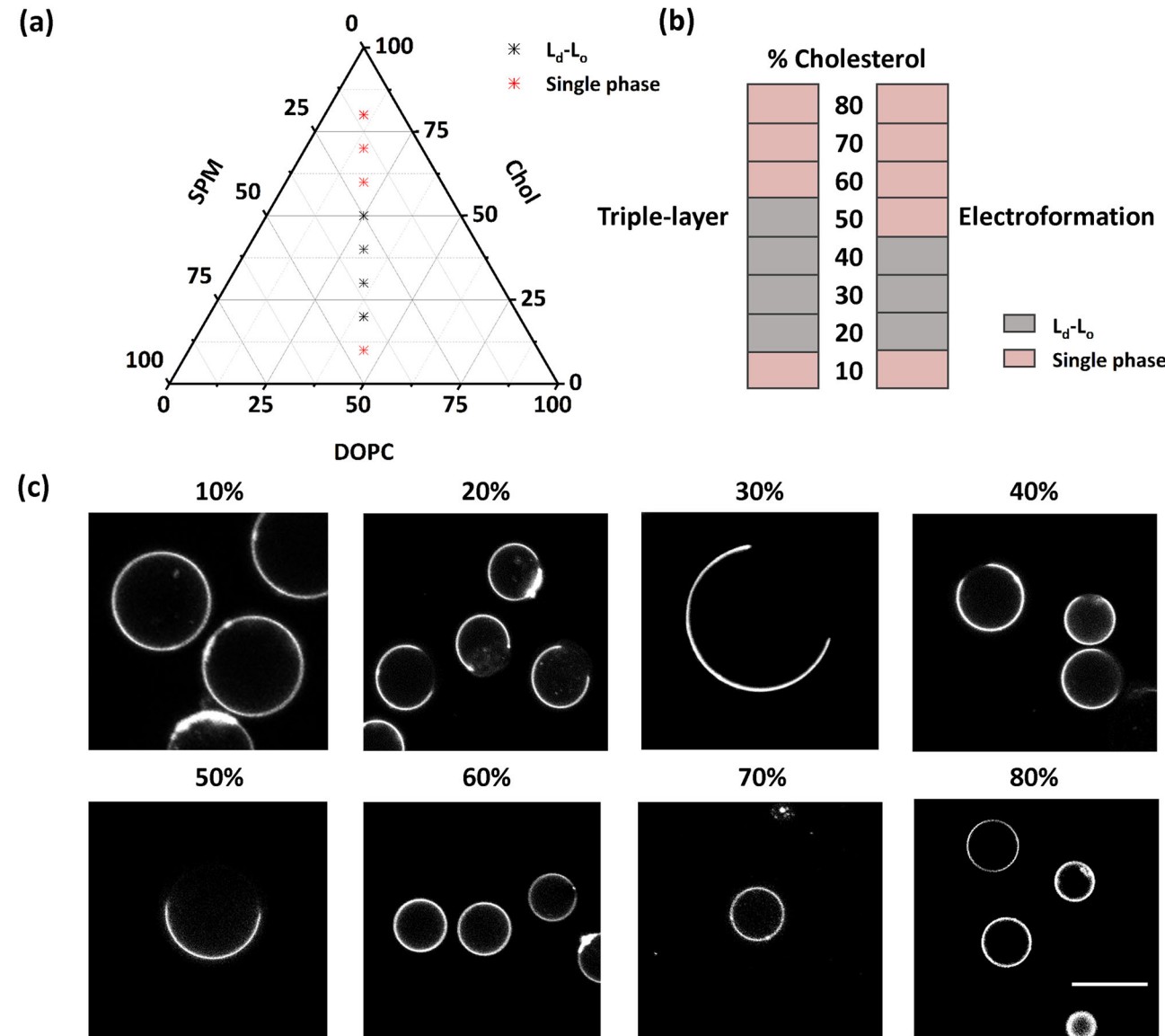

**Fig. 4 | Triple-layer inverted emulsion production of phase-separated GUVs.**
**a** Phase diagram of three-component lipid bilayer showing two-phase and single-phase regions produced from ternary mixtures with varying concentrations of cholesterol. DOPC:Sphingomyelin were at a fixed 1:1 molar ratio, and in additional GUVs are labeled with 0.5 mol% DiD which partitions into the Ld phase. Black dots represent the concentrations which resulted in co-existing phases (a vesicle population with more than 50% of the GUVs exhibiting domains), and red dots represent the concentrations that resulted in a single-phase (less than 50% with visible domains). **b** Comparing the percentage of phase separation amongst the GUVs produced with different cholesterol concentrations (DOPC:SPM – 1:1) between the triple-layer method and the electroformation (EF) method. **c** Confocal images (DiD signal) of phase-separated GUVs produced using the triple-layer method with cholesterol percentages of 10%, 20%, 30%, 40%, 50%, 60%, 70% and 80%. $n \geq 70$, data is collected from three or more independent experiments. Scale bars correspond to 20 μm.

Fig. 6). The same can be said of other reports in which membrane nanotubes were pulled from GUVs[53]. Previously, it has been observed that GM1 asymmetry in non-phase-separated GUVs (10% GM1) resulted in high membrane positive curvature which generated membrane nanotubes[54,55]. Indeed, we also observed outward curvature, but with only 1% GM1 and with a phase-separating membrane composition (Fig. 5c and Supplementary Video 6). This pronounced curvature with less GM1 compared to previous work could be due to the enrichment of GM1 to the Lo domain. Furthermore, the protrusions remain for over 6 h (Supplementary Video 7), suggesting that there is no flipping of GM1 or biotinylated lipids for long time periods. This opens the possibly of using such morphologies as polarized synthetic cells or anisotropic novel drug carrier systems. Thus, the outward-budding of the Lo domain is driven by two synergetic mechanisms. First, the GM1-enriched outer leaflet leads to a positive spontaneous curvature of this domain, arising from the large head groups of the GM1 glycolipids. Second, the domain boundary between the Lo- and the Ld-domain has a positive line tension. Therefore, the free energy of the Lo-domain is reduced when the budding decreases the length of the domain boundary[48].

To establish control of synthetic cell morphology, we modulated the protein concentration in one domain whilst keeping the other one constant. To do this, we systematically increased the concentration of streptavidin (10 nM, 20 nM, 40 nM, and 80 nM) inside the vesicle while maintaining a constant CT-B Al-594 concentration outside. As shown in Fig. 6, at equilibrium (after an hour), the pronounced membrane curvature in the Lo domain (red) was progressively reduced as the concentration of internal bound streptavidin in the Ld domain (green)

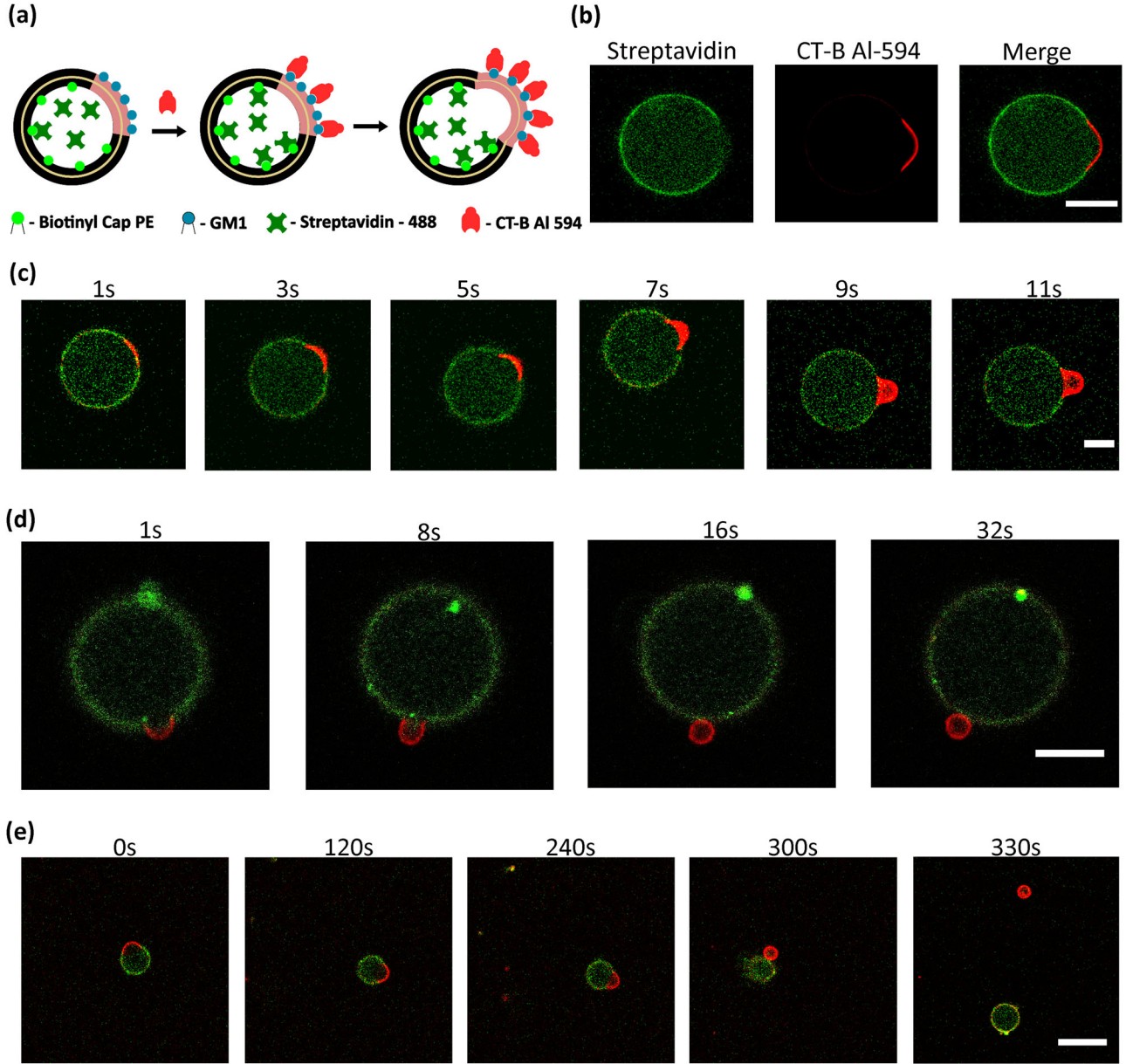

**Fig. 5 | Asymmetric phase-separated synthetic cells with sphingomyelin using the triple-layer inverted-emulsion method. a** Schematic representation of the asymmetric phase separated GUVs with biotinylated lipids (light green) binding streptavidin (green) in the inner leaflet of the Ld domain (black region) only, and GM1 (blue) binding CT-B Al-594 (red) in the outer leaflet of the Lo domain (pink region) only. **b** Confocal images of asymmetric phase-separated GUVs (50 mol% Chol, DOPC:SPM – 1:1, 1 mol% GM1 in the outer leaflet and 50 mol% Chol and DOPC:SPM – 1:1, 0.2 mol% biotinyl cap PE in the inner leaflet), showing streptavidin (green) binding to the liquid disordered domain, and CT-B Al-594 (red) binding to the Lo domain. Scale bars correspond to 20 μm. **c** Time-dependent curvature dynamics induced within the Lo domain of the asymmetric phase-separated vesicles after the addition of 20 nM CT-B Al-594. Scale bars correspond to 20 μm. **d** Time series of osmotic deflation induced budding-off of the curved CT-B AL-594 Lo domain (in red). Scale bar corresponds to 10 μm. **e** Time-series of osmotic deflation induced division of an Lo daughter vesicle from the Ld domain. Data is collected from three or more independent experiments. Scale bar corresponds to 20 μm.

increased ($n \geq 15$). This modulation of the protein concentration at the inner Ld leaflet only, counteracted the Lo domain curvature induced by CT-B Al-594 binding. At streptavidin concentrations of 40 nM and above, the vesicle's morphology is restored to a state close to its original spherical configuration (i.e., without CT-B).

This phenomenon mirrors the mechanism by which the outward budding of the Lo domain reduces the line free energy at the domain boundary when CT-B Al-594 is added. Therefore, we demonstrate the ability to control membrane curvature in one domain by modulating protein binding in another within a complex asymmetric phase-separated system. These findings suggest that such asymmetric systems offer a novel approach for studying the dynamic interplay between membrane domains within the same vesicle, enabling unprecedented control over domain-specific membrane curvature.

## Shape analysis and budding dynamics of asymmetric phase-separated GUVs

Next, we applied shape analysis to better understand the dynamic morphologies observed in our asymmetric phase-separated synthetic cells. This was done by minimizing the elastic energy of the two-domain vesicles in Fig. 5 as given by Eq. (S6) (see detailed discussion in the SI). Taking into account the constraints on the vesicle volume and

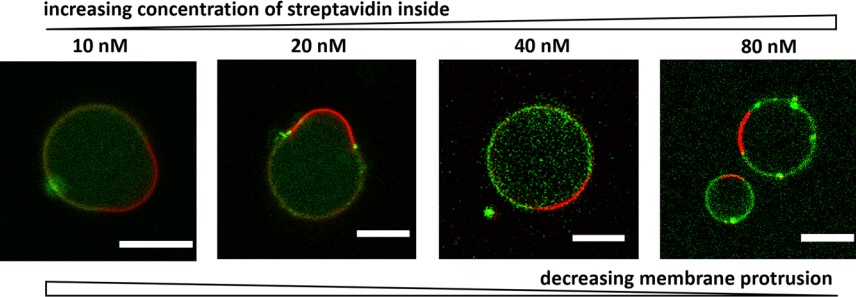

**Fig. 6 | Controlling synthetic cell morphologies by modulating the protein concentration.** Exemplary GUVs when increasing the initial concentration of Streptavidin protein (from left to right) in the lumen of the vesicles (prepared from four separate experiments, $n \geq 15$) results in reduced to no membrane protrusion of the Lo domain after Cholera toxin addition. $n \geq 15$, data is collected from three or more independent experiments. Scale bar corresponds to 10 μm.

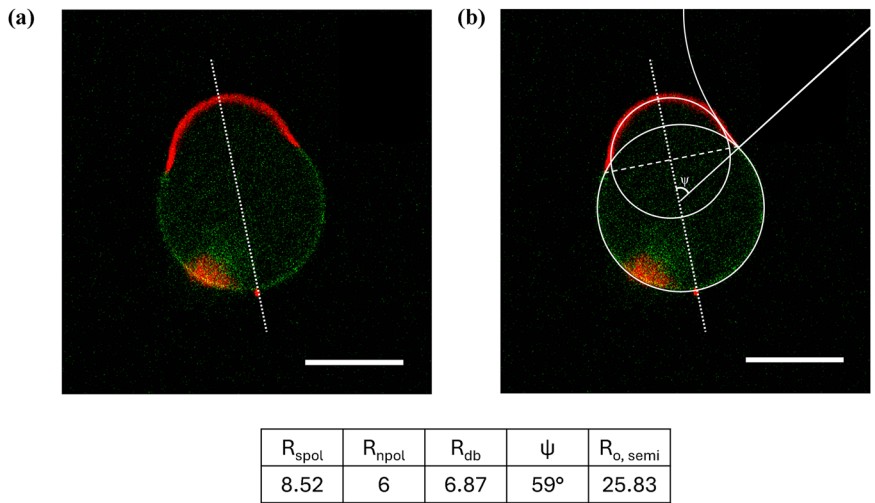

| $R_{spol}$ | $R_{npol}$ | $R_{db}$ | $\psi$ | $R_{o, semi}$ |
|---|---|---|---|---|
| 8.52 | 6 | 6.87 | 59° | 25.83 |

**Fig. 7 | Shape contour analysis for an axisymmetric two-domain vesicle. a** Confocal cross-section of vesicle with Lo domain (red) and Ld domain (green). The intersections of the axis of rotational symmetry (dotted line) with the domains define the vesicle's south and north pole. **b** The osculating circles that touch the shape contour at the north and south pole have the radii $R_{npol} = 6$ μm and $R_{spol} = 8.52$ μm, respectively. The radius of the domain boundary is equal to $R_{db} = 6.87$ μm, half the length of the dashed line perpendicular to the symmetry axis. The tilt angle $\psi = 59°$ is the angle between the symmetry axis and the membrane's normal vector at the domain boundary. The osculating semicircle (upper right corner) with radius $R_{o,semi} = 25.8$ μm touches the Lo domain at the domain boundary. $n = 10$, data is collected from three or more independent experiments. Scale bars: 10 μm.

surface areas, one has to minimize the shape functional given by Eq. (S7) for fixed volume $V$ and fixed surface areas $A_o$ and $A_d$ of the Lo and Ld domains. The elastic energy of the two-domain vesicle includes the bending energies of the two membrane domains which depend on the bending rigidities and the spontaneous curvatures of these domains. The bending rigidity of the Lo domain is larger than the bending rigidity of the Ld domain, which can be understood by the increased bilayer thickness of the Lo domain[56–58]. In addition to the difference in bending rigidity, protein asymmetry in our systems changes the spontaneous curvatures of the two domains. In general, the spontaneous curvature arising from protein asymmetry is unable to induce the formation of a closed neck for fixed vesicle volume but such a neck can be formed during osmotic deflation[52]. Therefore, we used deflation to achieve neck closure as in Fig. 5d and Supplementary Fig. 7a, similar to previous studies[52,59–61]. Together with the Supplementary Video 8, we conclude that osmotic deflation can also promote domain-induced budding in asymmetric GUVs, leading to the formation of two spherical membrane domains connected by a closed neck. In general, domain-induced budding is facilitated by the line tension of the domain boundary and by a large spontaneous curvature of the domain that forms the bud[51,52,59]. Furthermore, we also observed neck fission and, thus, vesicle division (see Fig. 5e and Supplementary Video 9). In the Supplementary Fig. 7b, the Lo buds have a bright red ring (due to

CT-B Al-594 binding) and are devoid of streptavidin (green) binding. The boundaries between the Ld and Lo membrane domains contribute the term $\lambda L_{db}$ to the free energy of the phase-separated membrane, which is proportional to the line tension $\lambda$ and to the total length $L_{db}$ of the boundaries. The line tension $\lambda$ is always positive, which implies that the membrane's free energy can be reduced by domain coarsening, i.e., by fusing two or more domains into one larger domain. This coarsening process eventually leads to only two domains and one domain boundary (as observed here). The additional asymmetric binding of proteins to specific membrane domains, changes their mean curvatures, with a more pronounced change of the Lo domain as displayed in Fig. 5b–d.

The observed two-domain vesicles are axisymmetric which implies that these shapes can be determined by their contours as illustrated in Fig. 7. The axis of rotational symmetry corresponds to the dotted line, which intersects the Lo domain at the north pole and the Ld domain at the south pole of the vesicle. The mean curvature of the vesicle membrane at the two poles is obtained by fitting two osculating circles to the shape contour, one at the north pole and another one at the south pole (see Fig. 7b). In general, an osculating circle is tangent to the curve, and has the same curvature at the point of contact. The radii of these two circles, denoted by $R_{npol}$ and $R_{spol}$, determine the mean curvatures $M_{o,npol}$ and $M_{d,spol}$ of the vesicle at the north (Lo) and south

pole (Ld) according to $M_{o,npol} = \frac{1}{R_{npol}}$ and $M_{d,spol} = \frac{1}{R_{spol}}$. Inspection of Fig. 7b reveals that the osculating circle at the south pole provides a good fit to the whole shape contour of the Ld domain. As a consequence, the Ld domain forms a spherical cap with constant mean curvature $M_d$,

$$M_d = M_{d,spol} = \frac{1}{R_{spol}} = M_{d,db} \qquad (1)$$

where $M_{d,db}$ represents the mean curvature of the Ld domain at the domain boundary. For the two-domain vesicle in Fig. 7, we obtain $M_{d,db} = \frac{1}{R_{spol}} = 0.117 \, \mu m^{-1}$.

In contrast, the osculating circle at the north pole deviates from the contour of the Lo domain close to the domain boundary which implies that the mean curvature $M_{o,db}$ of the Lo domain at the domain boundary is different from $1/R_{npol}$. In order to determine the mean curvature $M_{o,db}$, we define it in terms of its two principal curvatures $C_{1,o,db}$ and $C_{2,o,db}$. The first principal curvature $C_{1,o,db}$ of the Lo segment at the domain boundary is equal to,

$$C_{1,o,db} = -\frac{1}{R_{o,semi}} \qquad (2)$$

where $R_{o,semi}$ is the radius of the osculating semicircle that touches the Lo segment at the domain boundary (see upper right corner of Fig. 7b). The first principal curvature $C_{1,o,db}$ is negative because the tilt angle between the normal vector of the Lo segment and the symmetry axis decreases as we move along the semi-circle towards the domain boundary and because the first principal curvature is, in general, equal to the first derivative of the tilt angle with respect to arc length. The second principal curvature $C_{2,o,db}$ has the value,

$$C_{2,o,db} = \frac{\sin \psi}{R_{db}} = C_{2,d,db} = \frac{1}{R_{spol}} \qquad (3)$$

where the second equality follows from the continuity of the second principal curvature for an axially symmetric two-domain vesicle across the domain boundary. The mean curvature $M_{o,db}$ of the red Lo segment at the domain boundary is then given by,

$$M_{o,db} = \frac{1}{2}\left(C_{1,o,db} + C_{2,o,db}\right) = \frac{1}{2}\left(-\frac{1}{R_{o,semi}} + \frac{1}{R_{spol}}\right). \qquad (4)$$

For the two-domain vesicle in Fig. 7, we obtain the mean curvature $M_{o,db} = 0.0393 \, \mu m^{-1}$ as we approach the domain boundary from the Lo domain. Comparison with the value $M_{d,db} = \frac{1}{R_{spol}} = 0.117 \, \mu m^{-1}$ for the Ld domain at the domain boundary shows that the mean curvature is discontinuous across the domain boundary[62], a discontinuity that has not been elucidated in previous experiments. We also found such a mean curvature discontinuity for all other GUVs that were analyzed using the same method (Supplementary Fig. 8). The corresponding data obtained after fitting the vesicles (refer to Methods section, Vesicle shape contour analysis and Supplementary CAD file) are provided in Supplementary Table 5 of the Supplementary Information.

Figure 5d, e show examples of the bud formation and hence closed membrane necks between the two domains (Supplementary Fig. 9). As explained by Eq. (S8) in the SI, the domain boundary between the two domains moves out of the closing neck when the two domains differ in their Gaussian curvature moduli. The Gaussian curvature modulus $\kappa_G$ is expected to be negative with an absolute value that is comparable to the bending rigidity $\kappa$, that is, $\kappa_G \simeq -\kappa$ [60,62]. Because the Lo domain is more rigid than the Ld domain with $\kappa_o > \kappa_d$, the Gaussian curvature modulus $\kappa_{Go} \simeq -\kappa_o$ of the Lo domain is more negative than the Gaussian curvature modulus $\kappa_{Gd} \simeq -\kappa_d$ of the Ld domain. The energy term in Eq. (S5) then implies that the neck is

formed by the more flexible Ld domain[52], even though the associated shift of the domain boundary away from the neck could not be resolved because of the limited optical resolution of about 300 nm. Nevertheless, the conclusion that the neck is formed by the Ld domain has several interesting consequences. First, these necks can be characterized by their effective mean curvatures $M_{ij}^{eff}$ as defined in Eq. (S10). Second, the formation of the closed necks in Fig. 5d, e implies that the spontaneous curvature $m_d$ of the Ld domain exceeds the effective neck curvature $M_{ij}^{eff}$ as in Eq. (S11). Shape analysis leads to the values $M_{ij}^{eff} = 0.336 \, \mu m^{-1}$ for Fig. 5d and $M_{ij}^{eff} = 0.308 \, \mu m^{-1}$ for Fig. 5e. The same analysis has been applied to other vesicles with closed necks, leading to the numerical values in the Supplementary Table 6. Third, because the closed neck of the asymmetric two-domain vesicle in Fig. 5d, e undergoes fission we can also conclude that, in this case, the constriction force $f$ acting against this neck exceeds 20 pN as follows from Eq. (S13) and previous experiments[60].

## Discussion

To build complex synthetic cells or to directly mimic a eukaryotic cell, it is imperative to reflect its defining features. Here, we have presented a method to produce asymmetric lipid bilayers also exhibiting lateral membrane phase separation. This addresses the issue that most experimental techniques only produce symmetric membranes and even lack the capacity to encapsulate large biomolecules. Furthermore, existing techniques that are able to produce such complex combinations of lipid membranes such as hemifusion-based[18] and methyl-cyclodextrin-based[19] are limited to outer leaflet modifications, are restricted to specific lipids, require external agents such as fusogens, and require long preparation times. Our method, on the other hand, not only addresses these drawbacks but also provides three important advantages over existing techniques for synthetic cell production: (i) generating lipid membranes with a high degree of asymmetry and no restriction on lipid type, (ii) reproducible production of phase separated GUVs using the inverted emulsion method, and (iii) the combination of lipid asymmetry and phase separation in the same GUV. These unique advantages allowed us to assemble and study the consequences of asymmetric protein binding on asymmetric membranes, i.e., stable and controllable spontaneous membrane curvatures resulting in membrane budding. In addition, these budded regions could be further increased to induce membrane neck closure and complete fission of our synthetic cells.

All this is achieved through altering the layers and types of lipidic solvents involved in the inverted emulsion method, we can reproducibly generate both asymmetric GUVs as well as phase-separated ones. This is made possible because of the usage of squalene (for monolayer formation) as one of the lipid oils along with mineral oil (for W/O emulsion generation). The density of squalene is slightly higher than the density of mineral oil, therefore, partitioning of the two different compositions of lipids across these two solvents is not spontaneous. However, employing squalene as the sole oil phase for the entire inverted emulsion process is not a viable strategy. As we have previously demonstrated, GUVs generated from a single squalene-oil phase exhibit very low formation efficiency and suffer significant leakage of their aqueous contents due to the required high centrifugation speeds (>400 × $g$)[33]. This instability fundamentally undermines the goal of creating and maintaining a well-defined lipid asymmetry. To overcome these established limitations, we engineered the current oil system. Mineral oil is used for the initial droplet encapsulation, and as a physical barrier to prevent lipid mixing (also serving as a wash step to remove excess, non-incorporated dye). By reserving squalene for the destination phase, we decouple the requirements for stable emulsion formation from the final membrane composition. This design allows us to leverage the unique benefits of squalene for inducing phase separation (i.e., with cholesterol) without the associated penalties of poor yield and vesicle instability, thus

enabling the reliable production of intact, asymmetric as well as phase-separated GUVs. Furthermore, we implemented a spacer oil (between squalene and mineral oil) that is without lipids to further avoid lipid mixing and ensure a high degree of asymmetry. Bilayer asymmetry is proven via quenching experiments, first with simple lipid mixtures containing 0.5% of PE-NBD as the asymmetric lipid and later with complex lipid mixtures containing up to 1% of large head group lipids as the asymmetric lipid. Even with experimental observations lasting up to 6 h, the observed asymmetry and the resulting membrane curvature remained stable for the entire duration (Supplementary Fig. 10). Although we haven't tested higher concentrations of lipid asymmetry, up to 10 mol% in asymmetric lipid concentration within vesicles showed similar percentage of asymmetry in previous work[43]. However, it would be worthwhile to study asymmetric lipid concentrations beyond 10% to achieve higher relevance to eukaryotic membranes.

The biomimetic relevance of the GUVs was demonstrated by functionalizing specific leaflets with specific proteins (i.e., streptavidin, GFP, and CT-B Al-594). The results further prove a high degree of membrane asymmetry while retaining functional properties of the proteins during the production process which is crucial for synthetic cell development, that hasn't been addressed previously.

Finally, we have shown that the triple-layer method is able to successfully and reproducibly produce phase-separated asymmetric GUVs. Thanks to usage of squalene, cholesterol incorporation is efficient compared to previous methods reported, when only mineral oil was used (Supplementary Table 4). This is due to the fact that mineral oil contains more hydrocarbon chains that are saturated compared to squalene. Cholesterol is known to have a lower solubility in unsaturated oils which results in a higher partitioning towards the lipids formed at the oil-water interfaces with squalene[63]. These unique synthetic cells with complex membranes were able to exhibit cell-like behavior, with budding and division events, from a purely biophysical origin with no chemomechanical coupling of membrane fission to chemical reactions such as ATP or GTP hydrolysis. To assess the consistency and tunability of membrane curvatures, we analysed a set of ten GUVs using shape contour fitting (Supplementary Table 5) and sixteen vesicles undergoing neck closure (Supplementary Table 6). Across all GUVs, curvature discontinuities at the domain boundary were consistently observed. Our shape analysis further revealed a significant discontinuity in mean curvature across the domain boundary, quantitatively expressed as an average difference of $0.87 \pm 0.60\ \mu m^{-1}$ (Supplementary Table 5). This variability underscores the influence of local geometric factors (varying radii of Lo and Ld segments within the population) on membrane curvature. This unique observation is most likely a consequence of the protein asymmetry confined to specific domains which our method is able to produce. Further analysis also revealed the effective mean curvatures $M_{ij}^{eff}$ of the closed necks, which connect two spherical membrane segments with positive mean curvature. Such closed necks are only possible if the spontaneous curvatures of the two intramembrane domains exceed certain threshold values[64]. Analysis of vesicles undergoing budding and division showed a mean effective neck curvature of $0.307 \pm 0.034\ \mu m^{-1}$ (Supplementary Table 6), demonstrating a consistent curvature within the population. These results demonstrate the reproducibility and controllability of curvature generation in our system. The shape analysis conducted is based on the theory of curvature elasticity as reviewed[62] and extended in ref. 64. The basic ingredients of this theory are described in the SI, Eqs. S1–S8. In addition, this theory is applied to curvature discontinuity at domain boundary and to closed membrane neck configurations. Inspection of these equations shows that the vesicle shapes depend, in general, on the bending rigidity $\kappa_o$ and the spontaneous curvature $m_o$ of the Lo domain, on the corresponding curvature-elastic parameters $\kappa_d$ and $m_d$ of the Ld domain, as well as on the line tension $\lambda$ of the domain boundary. To obtain a quantitative comparison between theory and experiments, one would

have to measure these five membrane-elastic parameters, which is beyond the scope of the present work. On the other hand, our analysis demonstrates that the mean curvature of the membrane is discontinuous across the domain boundary, a generic morphological feature that has not been addressed in previous studies. Furthermore, our analysis also provides a lower bound of 20 pN for the constriction force, $f$, acting against the closed membrane neck.

A comprehensive study including a quenching assay, peripheral protein binding assay, protein encapsulation, phase separation with complex ternary mixtures, asymmetry, and shape analysis has not been attempted before. This work therefore unlocks the true potential of the inverted emulsion method for producing complex synthetic cells. With this technique, which can produce asymmetric bilayers as well as laterally phase-separated membranes, it is possible to obtain more accurate plasma membrane and organelle models, as well as constructing functioning synthetic eukaryotic cells from the bottom-up.

## Methods

### Materials
1-palmitoyl-2-oleoyl-glycero-3-phosphocholine (POPC), 1,2-dioleoyl-sn-glycero-3-phosphocholine (DOPC), Sphingomyelin (SPM), Cholesterol (Chol), 1,2-dioleoyl-sn-glycero-3-[(N-(5-amino-1-carboxypentyl) iminodiacetic acid)succinyl] (nickel salt) (DGS-Ni-NTA), GM1 Ganglioside (Brain, Ovine-Sodium Salt) (GM1), 1,2-dioleoyl-sn-glycero-3-phosphoethanolamine-N-(cap biotinyl) (sodium salt) (biotinyl Cap PE), and L-α-Phosphatidylethanolamine-N-(7-nitro-2-1,3-benzoxadiazol-4-yl) (Ammonium Salt) (NBD-DOPE) are purchased from Avanti polar lipids. Cholera Toxin Subunit B (Recombinant), Alexa Fluor ™ 594 Conjugate (CT-B Al-594), 1,1'-Dioctadecyl-3,3,3 ', 3'-Tetramethylindodicarbocyanine, 4-Chlorobenzenesulfonate Salt (DiD), Streptavidin Alexa-488, and Corning™ Glass Bottom 96-well microplates for fluorescence-based assays were all purchased from Thermo Fischer scientific. Chloroform, Beta-casein, mineral oil, squalene, glucose, sucrose, Tris-(hydroxymethyl)-amino methane (Tris), and sodium chloride (NaCl) were purchased from Sigma Aldrich. All chemicals were used as purchased unless otherwise stated. MilliQ® water is used for making all solutions.

### Inverted emulsion method
A desired lipid mixture (see Supplementary Tables 2 and 3, for various lipid compositions) in chloroform (400 μM) is dried in a glass vial using argon gas and further left in vacuum for 60 min. Following this, mineral oil is added to the same glass vial in the case of double layer method and mineral oil (for the inner leaflet lipid mixture) and squalene (for the outer leaflet lipid mixture), in the case of triple layer method. The vials are subjected to bath sonication in degassing mode for 15 min at room temperature. The final lipid-oil mixture, with a concentration of 400 μM, is stored at 4 °C and used within a week. First, 30 μL of mineral oil lipid-oil mixture (for double layer method) and squalene lipid-oil mixture (containing the outer leaflet composition) (for triple layer method) is added to β-casein coated (2 mg/mL for 30 min) microtiter plate (96-well) prefilled with 50 μL of desired outer solution (glucose or glucose + 10 mM Tris and 100 mM NaCl for experiments with proteins, -900 mM). At this stage, the formation of a lipid monolayer is allowed to take place for 30 min. Following this, 30 μL lipid-free mineral oil is added on top of the squalene lipid-oil mixture, in the case of triple layer method. Immediately, 50 μL of pre-fabricated W/O emulsion formed by rubbing an Eppendorf® tube containing inner aqueous solution (3 μL of sucrose or sucrose + 10 mM Tris and 100 mM NaCl buffer for experiments with proteins, -900 mM) in 200 μL of mineral oil on an Eppendorf® rack, three times, is added on top of the spacer lipid-free mineral oil (for triple layer method) and mineral oil (for double layer method). Finally, a centrifugation step is performed in swinging buckets at 200 × $g$ for 3 min. In the case of lipid mixtures for phase separation (to generate lipid domains), the

solutions are pre-heated, and the inverted emulsion method is performed at 60 °C. All the lipid compositions are presented in Supplementary Table 1 and Supplementary Table 2 in the supplementary information. At all instances, equi-osmolar solutions are used, unless otherwise stated.

### Electroformation

Electroformation experiments for the generation of phase separated GUVs were performed to compare with the population of phase separated GUVs produced using inverted emulsion methods. For this purpose, two indium tin oxide (ITO) coated plates are preheated to 90 °C and cooled to room temperature before applying 10 μL of lipid mixture (POPC: SPM – 1:1, Chol 10/20/30/40/50/60/70/80) in chloroform on the ITO coated side. The lipid-coated ITO plates were desiccated under vacuum for 45 min. A chamber was formed with the lipid coatings sides facing towards each other using a 2 mm thick hollow rectangular Teflon spacer (20 mm × 40 mm). Pre-heated 900 mM sucrose solution was injected into the chamber and sealed with metal clips. Using copper tapes, the ITO plates are connected to a function generator and placed inside a hot oven set at 60 °C. A sine wave function at 10 Hz and 2 Vpp was applied to the chamber for 3 h. Finally, the GUV suspension was collected by tapping and pipetting, and imaging was performed at room temperature. Measurements were taken from distinct samples where $N \geq 3$.

### Lipid mixing analysis

Using tracer lipids in the oil layers, we tried to visualize the extent of lipid mixing in both double-layer and triple-layer methods as well as triple-layer without spacer oil. For this purpose, initially 600 μL of 900 mM glucose solution is added to the Eppendorf® tubes, followed by pipetting 200 μL of NBD-DOPE loaded mineral oil or squalene over the glucose solution. In one of the conditions, spacer oil, mineral oil without lipids, is added on top of the squalene layer. In the next step, W/O droplets (900 mM sucrose solution) in DID containing mineral oil is added. Following this, the tubes are centrifuged at 1000 × g for 3 min in an Eppendorf® mini centrifuge (Centrifuge 5424 R). After the centrifugation step, the bottom-most oil layer, interfacing the glucose solution, was observed for any change in color (from yellow to bluish yellow).

### Quenching assay

After the production of NBD-tagged PE (0.5 mol%) containing vesicles using the above described double and triple layer inverted emulsion methods, a 10 mM final concentration of freshly prepared sodium dithionate is injected in the outer solution. This injection can be either done after removing the top oil layers or without it based on the requirements of the experiment. After the successful injection of the quencher, end-point fluorescence intensity measurements or time-lapse experiments were performed using confocal microscopy. Fluorescence intensity percentage is calculated via Eq. 1 using Fiji (ImageJ) software for vesicles with uniform membrane intensity before and after the addition of the quencher. Statistical significance between groups was assessed using a two-sided Mann–Whitney U test (OriginPro, Origin Labs software). The null hypothesis was that the median quenching percentages between the two groups compared (double-layer vs. triple-layer) were equal. The significance level was set at $p < 0.01$. For the 'Asymmetric NBD Outside' condition, the difference between the double-layer and triple-layer methods was statistically significant ($p < 0.0001$), leading to the rejection of the null hypothesis, lack of a significant difference ($p > 0.01$) is denoted with n.s.

$$\text{Fluorescence Intensity (\%)} = \frac{\text{Intensity}_{\text{before quenching}} - \text{Intensity}_{\text{after quenching}}}{\text{Intensity}_{\text{before quenching}}} \times 100 \quad (5)$$

### Protein-lipid asymmetry experiments

For experiments with proteins encapsulated within the lumen of the vesicle, a streptavidin concentration of 10 nM, 20 nM, 40 nM or 80 nM was used to bind biotinylated lipids present in the membrane. In this case, the protein solution is mixed with an inner solution that is used to make W/O emulsions. In all other cases, where the proteins are in the outer solution, the desired concentration of protein solution (final concentrations, GFP 100 nM, streptavidin 20 nM, and cholera toxin subunit B (CT-B Al-647) 20 nM) with matching osmolality is injected into the outer solution after the formation of the GUVs. For osmolarity induced bud division experiments, 2.5 μL of 2 M glucose solution was injected into the outer solution creating a 50 mOsm hypertonic solution to deflate the GUVs, after cholera toxin binding. Time-lapse and end-point experiments were performed using confocal microscopy. Measurements were taken from distinct samples where $N \geq 3$.

### Microscopy

All the experimental results were obtained using a confocal fluorescence microscope (SP8, Leica). Vesicles with NBD-tagged lipids/streptavidin-alexa 488/GFP were excited with a laser excitation of 488 nm and the emission was detected between 498 and 540 nm. In the case of cholera toxin subunit-B tagged with Alexa Fluor-594, an excitation wavelength of 551 nm is used, and emission is collected between 565 and 620 nm. Finally, for DiD labeled vesicles, excitations are set at 638 nm and emissions were captured in the range of 648–710 nm. Sequential channel acquisition is performed whenever multiple fluorophores are employed. Acquired confocal images and videos are treated using LASX (Leica) or Fiji (ImageJ) software. Measurements were taken from distinct samples where $N \geq 3$.

### Vesicle shape contour analysis

For shape analysis, Autodesk® AutoCAD® was used to fit the vesicles. An exemplary CAD file for the GUV fitting method is included in the supplementary information. To perform the contour analysis, the GUV image file with scale bar is imported into the AutoCAD® file and the image is adjusted to match the dimensions of the GUVs to that of the CAD file via scaling (in the example file, the scale bar is 10 μm). To determine the axis of rotational symmetry for a two-domain GUV (Fig. 7a), two osculating circles fitting the Ld (green) and Lo (red) domains at the south and north pole are drawn with radii, $R_{spol}$ and $R_{npol}$, respectively (Fig. 7b). The line connecting the centers of the two circles represents the axis of rotational symmetry. The diameter $2R_{db}$ of the domain boundary is then obtained by drawing a line perpendicular to the rotational symmetry axis and connecting the two cross-sectional points of the domain boundary with the shape contour (where the Ld domain (green) meets the Lo domain (red)), shown as a dashed line in Fig. 7b. A semicircle is drawn with its corner at the domain boundary (left or right side of the GUV, in Fig. 7b, it is drawn on the right side) having a radius of $R_o$,semi, with the arc of the semicircle fitting the Lo domain contour (red) as shown in Fig. 7b and Supplementary Fig. 8. The base line of this semicircle is parallel to the membrane's normal vector at the domain boundary, which forms the tilt angle $\psi$ with the rotational symmetry axis (Fig. 7b). These geometric constructions are used to calculate the parameters for the equations mentioned in the main text as well as in the SI and presented in Supplementary Table 5.

Similarly, the effective mean curvatures ($M_{ij}^{eff}$) for vesicles with closed neck configurations are calculated by measuring the radii of the green region ($R_i$) and red region ($R_j$). The radii were measured by fitting a circle using circle function in AutoCAD® (an example fit can be found in the CAD file uploaded as in the supplementary information). Measurements were taken from distinct samples where $N \geq 3$.

**Reporting summary**

Further information on research design is available in the Nature Portfolio Reporting Summary linked to this article.

## Data availability

Source data is available for Fig. 2c, and Supplementary Figs 2, 3, 8, and 9 in the associated source file. Source data are provided with this paper.

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

## Acknowledgements
This work is part of the MaxSynBio consortium, which is jointly funded by the Federal Ministry of Education and Research of Germany and the Max Planck Society.

## Author contributions
T.R. and N.Y. conceived the project. T.S. and N.Y. performed the experiments. RL developed the theory. R.L. and N.Y. performed mathematical calculations. N.Y., T.S., R.L., and T.R. analyzed the data and wrote the manuscript.

## Funding

## Competing interests
The authors declare no competing interests.
