## [Transparent Peer Review file · Nature Communications]

Engineering synthetic cells with intramembrane domains possessing distinct bilayer asymmetries

Corresponding Author: Dr Tom Robinson

Version 0:

Reviewer comments:

Reviewer #1

(Remarks to the Author)

This manuscript by Yandrapalli et al. focuses on the production of asymmetric, phase separated vesicles prepared by emulsion phase transfer, where an alteration of the conventional method, employing a lipid-free oil spacer, is used to more controllably form asymmetric giant unilamellar vesicles, as determined by protein binding and fluorescence quenching of the vesicle membrane. The authors then use this method to produce phase separated vesicles, successfully segregating protein-binding lipids to the liquid disordered (Ld) and liquid ordered (Lo) phases respectively, and show that protein binding can be controllably localized to either face (and domain) of the phase separated vesicle. Finally, the budding off of the protein-bound Lo phase is monitored, and analysis of the shape and budding dynamics is evaluated.

This work represents an advance in the fabrication of lipid vesicle-based cell models by combining asymmetry and phase separation control to localize proteins to the membrane in a user-defined manner. This builds on work in the field that have shown (i) the production of asymmetric vesicles, including the author's prior work (10.1016/j.bpj.2022.12.017) and others which combine membrane asymmetry and protein binding to localize proteins to different faces of a vesicle membrane (<https://doi.org/10.1021/acssynbio.2c00564>), as well as (ii) the production of phase separated vesicles by emulsion phase transfer (10.1039/c7sc04309k).

As such, whilst the work is interesting, my main concern is in its novelty. The main focus of this paper is on vesicle characterisation, including their budding behavior, and for such work to be considered in Nature Communications (as opposed to a field-specific journal) I think the novelty of this manuscript needs to be strengthened. To do so I suggest the following additions;

1) Demonstrating novel functionality (beyond simply binding fluorescent proteins) of the asymmetric phase separated vesicles. Could a proof-of-concept experiment be performed which shows how such control of protein localization results in improved performance of the vesicles as synthetic cells, for example through adhesion, biosynthesis or other improved functionality?

2) Further investigation into the effects of protein localization and stability of such vesicles. The budding, whilst interesting, could be seen as a disadvantage with respect to synthetic cell stability, and it would be interesting to further investigate this. If proteins are localized to both faces of the Lo domain, can this budding be prevented during vesicle deflation? And how long does the protrusion of the asymmetric, phase-separated GUVs persist? This is very interesting from the perspective of lipid flip-flop kinetics, where the presence of the domain boundary may potentially affect flip-flop rates compared to a single phase asymmetric membrane.

I also have some suggestions and questions regarding the experimental approach, which should be explored to better validate the new method presented in the manuscript;

3) The authors present the triple-layer inverted emulsion method as an improvement on prior work. The use of a spacer oil is interesting, and I can see how the use of oils of differing density could be used to better separate the lipid-containing squalene layer (outer leaflet) and W/O mineral oil emulsion (inner leaflet). The authors state that there is no lipid mixing during the production process, but there is no data that studies this. It would be interesting to observe whether lipid mixing occurs (perhaps through the use of fluorescent tracer lipids in the squalene and mineral oil emulsion phases), with and

without the mineral oil spacer oil. This would facilitate a better understanding of the role of the mineral oil spacer, and how necessary it is in the formation of the asymmetric vesicles.

4) In Figure 1C, the authors show fluorescence quenching data for double-layer and triple-layer emulsion phase separated vesicles. The authors include convincing data demonstrating the asymmetry of GUVs produced via the triple-layer method, but only produce symmetric vesicles using the double-layer method. The authors cite their prior work (10.1016/j.bpj.2022.12.017) and say that the triple-layer method gave better results in this study, than the double-layer method in the previous study. However, I believe the membrane asymmetry is different, with the focus of the prior work the asymmetric incorporation of NBD-labelled bacterial lipopolysaccharides. The authors should perform fluorescence quenching on GUVs prepared via the double-layer method using the NBD-DOPE lipid used in this work, and the fluorescent quenching compared for double- and triple-layered. This, combined with 3) above will enable a better evaluation for the use of the triple-layer phase transfer method as compared to the simpler double-layer method.

5) The authors report improved cholesterol integration within phase-separated GUVs due to poorer solubility of the molecule in squalene. Has this been reported previously using the double-layer method? Is the cholesterol present in the squalene layer of the assembled phase column only, or also in the mineral oil emulsion that is layered on top?

6) Could the authors provide more data for the curvature of the L_0 phase for i) asymmetric phase separated vesicles and ii) the symmetric phase separated vesicles after Ct-B AL 594 binding? In each case only one GUV is presented (asymmetric Fig 4C, symmetric Fig S3). For Fig S3, in the overlay of the image the domains do not align (probably due to timelapse between taking measurements in different fluorescence channels). Providing images of more GUVs would enable the reader to better evaluate the range of curvatures shown for asymmetric/symmetric phase separated GUVs respectively.

7) The shape analysis of vesicle curvature is interesting, but is only applied to a small number of asymmetric GUVs (for curvature discontinuity, 6 GUVs are evaluated and for the closed membrane necks 10 GUVs are evaluated). Were these GUVs prepared from a single phase transfer experiment? It would be good to apply this method to a larger number of GUVs, to better understand the range of curvatures observed the frequency with which the protrusion of the L_0 domain occurred across a vesicle population.

Finally, the methods section needs to be improved to enable reproducibility of the work;

8) For osmolarity-induced bud division experiments, the final solution osmolarity is not described (only that 2.5 μ l of 2M glucose solution is added). Please amend this so it is clear to the reader the final external solution osmolarity (and the osmotic gradient across the membrane).

9) The description of shape analysis in the method is minimally described - the only detail is that Autodesk AutoCAD was used to fit the vesicles. Please give sufficient information for a reader to reproduce the shape analysis.

10) How are the mathematical calculations performed? Please give details.

Reviewer #2

(Remarks to the Author)

This is a very interesting manuscript. It's well written, details are clear, data presentation is very logical, and conclusions are generally well justified.

The curvature formalism capable of considering the lipid mixture and effects of lipid ordered and lipid disordered phase is appreciably rigorous, and I haven't seen that applied in the field of synthetic cell engineering before. Authors used more biologically relevant lipids, which increases the value of this work. Authors also interestingly claim better incorporation of cholesterol.

Addressing few issues would improve this manuscript:

The authors mention that others have proffered methods for preparing asymmetric membranes, and the manuscript compares double layer emulsion to triple layer emulsion in figure 1. However, the authors did not compare the ability to form asymmetric bilayers between their method (triple layer emulsion) and double layer emulsion.

Moreover, authors significantly modify the lipid compositions and emulsion formulation between published double layer emulsions and what they test as double layer here. That they didn't overtly compare asymmetry between the double and triple layer methods is odd too because they validate that their assay would work with both methods.

Authors were not adequately specific about how successful they were with cholesterol incorporation beyond saying that they were able to see changes in phase separation dynamics with a range of working concentrations during the preparation.

Interesting methods: The spacer oil that separates the two lipid-in-oil solutions that contain the intended inner and outer leaflet lipid compositions,

Strictly speaking Kruskal-Wallis should never be conflated with ANOVA, also I want more explanation as to why the authors used something really exotic here.

They chose a nonparametric multi category test, if nonparametric due to dynamics of liposome formation, why not just Mann-Whitney with Bonferroni correction.

There should be justification for nonparametric tests (even though they are always more stringent than parametric tests) and

explanation for the choice of Kruskal Wallis over the more common Mann-Whitney.

The authors say that cholesterol is more soluble in mineral oil than in squalene, which I am inclined to believe, but the manuscript don't give citation or explanation as to why (the explanation to me would be that mineral oil has cyclic hydrocarbons and aromatics).

Likewise there are many small molecule quenchers used, I would like a citation or quick mechanistic comment about the sodium dithionate.

Overall, it's a very interesting paper. I believe after fixing and clarifying the above described issues, it will be a valuable publication of interest to our community.

Reviewer #3

(Remarks to the Author)

The manuscript by Yandrapalli et al. describes the development of an emulsion phase transfer method to generate giant unilamellar vesicles with an asymmetric lipid bilayer. This is an interesting concept but I have a number of concerns that need to be address, especially regarding the controls used to quantify if the method is an improvement over existing procedures.

1) Page 3. In the introduction, the authors state that "In contrast, the outer leaflet is devoid of such negatively charged lipids". While they are correct PS is not present in the outer membrane of healthy cells, the outer membrane is still negatively charged using alternative lipids. If this were not the case, drug delivery with positively charged polymers/lipids would not work.

2) Page 4. The authors have not included droplet interface bilayers as other methods to create asymmetric bilayers.

3) Page 4/5. There needs to be more discussion of previous methods to produce asymmetric bilayers and their drawbacks/differences to what the authors have developed. For instance, the sentence "Presently, to our knowledge, there is no methodology that can produce asymmetric membranes, along with the complexity of phase separation that can also sustain protein functionality." needs to be more thoroughly evidences and explained.

4) Page 7. The authors state "To achieve a high degree of bilayer asymmetry in GUVs, the challenge is to prevent mixing of the two different lipid containing oil solutions. Failure to do so will result in contamination of inner leaflet and outer leaflets with the opposite lipids and decrease the degree of asymmetry." This is not evidenced. Furthermore, these are the critical control experiments that are not performed throughout the paper. The control vesicles are prepared symmetrically, but this is not what the authors are claiming they are improving on. There needs to be controls for alternative state of the art asymmetric vesicle preps, for instance those generated by the authors themselves (doi: 10.1016/j.bpj.2022.12.017) and/or just adding an emulsion formed with the inner lipid to a phase transfer column prepared with the outer lipid. These controls are required throughout the work (especially Fig 1, 2, and 4) to actually back up the claims that they are improving on previous methods.

5) Page 9. Authors state "To avoid this, either the surface area of the interface between squalene and glucose solution has to be increased or the number of W/O droplets needs to be reduced." Can the lipid concentration be increased?

6) Page 10. Authors state "The lipid composition for this study was 99.5 mol% POPC and 0.5 mol% NBD-DOPE." Did the asymmetric quantification change when different % of the asymmetric lipid was used. I.e. only 0.5% is put on one side. This is the case in later experiments too, when only a small % is made asymmetric.

7) Page 11. Authors state "This suggests that a significant amount of the GUVs, 93 %, remained asymmetric for the duration of the experiment (approximately 1 hour) with no observable permeation or morphological changes." Did the authors measure past 1 hour? This seems like a very short time to quantify.

8) Fig 2. Why did the authors only test the single asymmetric membrane by adding the proteins from the outside? Looking at 2e/f the streptavidin signal is much weaker at the membrane this way (albeit with the other lipid on the outside too), is there a reason for this?

9) Page 20. Authors state "Both streptavidin (inside) and CT-B AL-594 (outside) proteins were used to further enhance the asymmetry and applicability of the system." Please explain this in more detail.

10) Page 20 and throughout. There needs to be a better way of explaining the lipid mix and which are in the outer/inner. For instance "DOPC:SPM:Chol - 1:1:2, 0.2 mol% biotinyl cap PE, 1 mol% GM1 and 0.5 mol% DiD" does not explain the system well enough.

11) Page 22. Authors state "In contrast, when adding CT-B AL-594 to symmetric phase-separated GUVs, no such protrusion was observed (Figure S3)." Can they explain why this might be? I.e. the presence of GM1 in the inner phase stops the protein from protruding the phase, but this is not obvious. Also on the same page there is another sentence that is not explained and difficult to understand "Note that there is no binding of streptavidin in the Lo domain (red) suggesting that there are no biotinylated lipids in the Lo phase, and that there is no large head group bearing lipids on the inner leaflet to compensate for the asymmetry of the GM1 in the outer leaflet unlike in symmetric phase-separated GUVs".

12) With regards to the curvatures quantified. There is mathematical modelling of individual vesicles, but no discussion or analysis of how they compare or how consistent they are. Or how these values change with altered composition/concentrations.

13) Page 29, authors state “These unique synthetic cells with complex membranes were able to exhibit cell-like behavior, with budding and division events, from a purely biophysical origin with no additional energy input.” Is this fair if the authors are adding protein and putting the vesicles under osmotic pressure to attain these features?

14) Methods. The authors use the phrase ‘briefly’ multiple times. The materials and methods should be exhaustive for someone to repeat. Regarding density of squalene vs mineral oil, this depends on the mineral you purchase, which is not given. There are multiple points where concentrations are not obvious or amounts added at steps are not given.

15) As is good practice, source data should be made available on a repository.

Version 1:

Reviewer comments:

Reviewer #1

(Remarks to the Author)

I was pleased to re-review this manuscript by Yandrapalli et al. The authors have taken on board reviewer comments and strengthened their manuscript, particularly regarding the control experiments necessary to verify the new triple-layer inverted emulsion phase transfer method. Further experiments were also conducted that demonstrate control of vesicle domain curvature through binding of proteins to differing domains and faces of the vesicle. These additions have largely addressed my concerns, and I believe the manuscript is suitable for publication in Nature Communications after addressing the following:

- I could not see information on the lipid mass and concentration in mineral oil / squalene in the inverted emulsion method section – please add to improve reproducibility.
- Figure 2C lists GFP as inside the GUV and the NTA lipid on the outside. I believe the opposite is the case? Please correct.
- Figure 3 – can you list the number of vesicles analysed for each chol% studied?
- Figure S4: Please show exemplar images for each of the compositions studied alongside the phase diagram.
- Figure S6 is a very interesting addition to the manuscript. Could you please list the number of vesicles studied for each streptavidin concentration? This is missing.
- “In the Supplementary Figure S4(b), the Lo buds have a bright red ring (due to CT-B AI-594 binding) and are devoid of streptavidin (green) binding” – should be Supplementary Figure S7?
- In the SI, in the “Shape parameters of two-domain vesicles” section, the final sentence ends stating there are five elastic shape parameters, but this sentence is not finished?
- Figure S8: 10 vesicles are presented but the caption states 6 are shown.
- Please check the naming of all equations in the manuscript and SI – I believe due to manuscript additions there are multiple equations mislabelled in the manuscript text.

Reviewer #2

(Remarks to the Author)

The authors addressed all my concerns and questions. I think the new data and explanations improved the manuscript. I have no more comments.

Reviewer #3

(Remarks to the Author)

I thank the authors for their extensive response and adding in the new data to the manuscript, although I still have a number of concerns and queries.

From my reading this manuscript mentions two main findings, that they produce higher degree of lipid bilayer asymmetry (1) and better phase separated asymmetric GUVs (2), compared to previous emulsion phase transfer methods.

For the first point, all reviewers asked for more comparison to previous methods, which the authors gave in the form of the double-layer method (Fig 1). Surprisingly, they only saw an improvement in the symmetry of the outside leaflet, which dramatically reduces their argument. Furthermore, the authors did not even quantify the comparison of the triple layer (with an oil free layer) to triple layer (without an oil free layer). They did this with a visual inspection (Fig S1), which did show the oil free version kept the two layers separated somewhat, even after centrifugation. They need to actually measure the asymmetry to see if the oil free layer is required. Furthermore, in my original review I said the authors should also add an emulsion formed with the inner lipid+mineral oil (with time for the monolayer to form) to a phase transfer column prepared with the outer lipid+squalene (again with time for the monolayer to form). This again is likely to keep the asymmetry to a

similar level and would be much simpler. The authors even state that “the rate of formation of lipid monolayer is much longer than the rate of droplet transfer”, therefore minimal lipid/oil mixing is likely not to cause a problem if the monolayers are already formed. Furthermore, there is little quantification of protein binding in Fig 2 (just an individual GUV for each condition in Fig S2), could the other control methods provide similar results here even if their asymmetric values measured using NBD aren't as good (is protein being less sensitive than the NBD assay)?

For the second point, the improved phase separation seems to come entirely from the improved ability to add cholesterol into the membrane (from my reading). The authors state this is because of the use of squalene. However, as like the first point, the squalene is not reliant on the triple layer with an oil-free layer. Therefore, can similar results for this part also be obtained using triple layer without the oil-free layer or incubating the monolayers separately?

Overall, you can see that my concern is that the system is over-engineered, and not all of the parts are required. Or that similar results can actually be obtained in a simpler way than careful layering of four layers together.

Other points:

For “Even with experimental observations lasting well beyond 6 h, the observed asymmetry and the resulting membrane curvature remained stable for the entire duration” the authors need to show this data.

The supplier or preparation of His-GFP is missing from the text.

Version 2:

Reviewer comments:

Reviewer #3

(Remarks to the Author)

All my final comments have been addressed. I support publication.

REVIEWER COMMENTS

Reviewer #1

This manuscript by Yandrapalli et al. focuses on the production of asymmetric, phase separated vesicles prepared by emulsion phase transfer, where an alteration of the conventional method, employing a lipid-free oil spacer, is used to more controllably form asymmetric giant unilamellar vesicles, as determined by protein binding and fluorescence quenching of the vesicle membrane. The authors then use this method to produce phase separated vesicles, successfully segregating protein-binding lipids to the liquid disordered (Ld) and liquid ordered (Lo) phases respectively, and show that protein binding can be proteins can be controllably localized to either face (and domain) of the phase separated vesicle. Finally, the budding off of the protein-bound Lo phase is monitored, and analysis of the shape and budding dynamics is evaluated.

This work represents an advance in the fabrication of lipid vesicle-based cell models by combining asymmetry and phase separation control to localize proteins to the membrane in a user-defined manner. This builds on work in the field that have shown (i) the production of asymmetric vesicles, including the author's prior work (10.1016/j.bpj.2022.12.017) and others which combine membrane asymmetry and protein binding to localize proteins to different faces of a vesicle membrane (<https://doi.org/10.1021/acssynbio.2c00564>), as well as (ii) the production of phase separated vesicles by emulsion phase transfer (10.1039/c7sc04309k).

As such, whilst the work is interesting, my main concern is in its novelty. The main focus of this paper is on vesicle characterisation, including their budding behavior, and for such work to be considered in Nature Communications (as opposed to a field-specific journal) I think the novelty of this manuscript needs to be strengthened. To do so I suggest the following additions:

1) Demonstrating novel functionality (beyond simply binding fluorescent proteins) of the asymmetric phase separated vesicles. Could a proof-of-concept experiment be performed which shows how such control of protein localization results in improved performance of the vesicles as synthetic cells, for example through adhesion, biosynthesis or other improved functionality?

We appreciate the reviewer's thoughtful feedback and suggestions. While we recognize the value of demonstrating additional functionality such as adhesion or biosynthesis, we believe the current focus on controlling protein localization (not only to a domain but also to a leaflet) and analysing the resulting budding dynamics already offers significant novelty as this has never been done before. Our work uniquely integrates lipid membrane asymmetry and liquid-liquid membrane phase separation to precisely localize proteins to specific lipid domains, a combination not previously achieved. This platform mimics key aspects of cellular membranes, particularly the asymmetric distribution of lipids and proteins, and allows for the controlled study of protein-induced membrane budding and shape changes. Relevant for bottom-up synthetic cell formation and a process relevant to biological vesicle formation and trafficking. While future studies can explore additional functionalities, the primary novelty here lies in our ability to controllably localize proteins and induce budding in phase-separated vesicles, which represents a substantial advancement in synthetic cell design and provides a robust foundation for future applications. We believe this contribution is sufficiently novel for Nature Communications.

We would like to highlight that we have not just shown binding of proteins, we have also shown controllable morphological (curvature) changes which were a direct result of the asymmetry binding. Furthermore, the curvature changes resulted in budding of smaller “daughter” GUVs. In the context of bottom-up synthetic cells, this is important to demonstrate a new mechanism for synthetic cell division. Note that both of these functionalities (morphological changes and budding off) were not seen in simple symmetry systems (i.e. controls).

To emphasize the points above, the following text has added to the discussion section:

Furthermore, existing techniques that are able produce complex combinations of lipid membranes such as hemifusion-based¹⁸ and methyl-cyclodextrin-based¹⁹ are limited to outer leaflet modifications, are restricted to specific lipids, require external agents such as fusogens, and require long preparation times. The latter point being unsuitable for membrane dynamic studies. Our method, on the other hand, not only addresses these drawbacks but also provides three important advantages over existing techniques for synthetic cell production: (i) generating lipid membranes with a high degree of asymmetry and no restriction on lipid type, (ii) reproducible production of phase separated GUVs using the inverted emulsion method, and (iii) the combination of lipid asymmetry and phase separation in the same GUV. These unique advantages allowed us to assemble and study the consequences of asymmetric protein binding on asymmetric membranes, i.e., stable and controllable spontaneous membrane curvatures resulting in membrane budding. In addition, these budded regions could be further increased to induce membrane neck closure and complete fission of our synthetic cells.

2) Further investigation into the effects of protein localization and stability of such vesicles. The budding, whilst interesting, could be seen as a disadvantage with respect to synthetic cell stability, and it would be interesting to further investigate this. If proteins are localized to both faces of the *Lo* domain, can this budding be prevented during vesicle deflation? And how long does the protrusion of the asymmetric, phase-separated GUVs persist? This is very interesting from the perspective of lipid flip-flop kinetics, where the presence of the domain boundary may potentially affect flip-flop rates compared to a single-phase asymmetric membrane.

Thank you for your insightful suggestions. We understand the importance of further investigating the stability of the asymmetric, phase-separated vesicles, particularly in relation to protein localization and its impact on budding.

To address your first point, investigating whether protein localization to both faces of the *Lo* domain can prevent budding during vesicle deflation is the logical next step. Such experiments could reveal how symmetric protein distribution across both leaflets impacts membrane stability, potentially counteracting forces that drive budding. Multiple works using proteins, and symmetric membranes have already been shown, cited in the manuscript as well (<https://doi.org/10.1021/acssynbio.2c00564>). However, your proposed experiment, i.e. investigating whether protein localization to both faces of the *Lo* domain can prevent budding during vesicle deflation, is counter intuitive. This is because, there will not be a bud to undergo fission in the first place as the membrane will be saturated with proteins on both sides of the membrane (as seen in Figure 2e). In our system, membrane curvature is induced by asymmetry alone, and not by symmetry (i.e. proteins localised to both *Lo* faces as suggested). We can also observe that in our systems where we have large head group lipids on both sides of the membrane bound to proteins, we see no protrusions – a condition similar to what you are suggesting. Furthermore, we have included a general explanation on why vesicles remain spherical and without any protrusions when the area is conserved, for more clarity.

We have included the following text in the SI:

Geometric constraint for shape transformations

The volume-to-area ratio v is restricted to the range $0 < v \leq 1$, where the maximal value $v = 1$ corresponds to a spherical vesicle shape. Thus, for a given surface area A of the vesicle membrane, the vesicle volume V cannot exceed its maximal value $V_{\max} = A^{3/2}/(6\sqrt{\pi})$. In general, we should distinguish the apparent area of the membrane from its true area because some excess area can be stored in the membrane's shape fluctuations. However, the vesicles analyzed here did not exhibit such fluctuations on the scales above 300 nm as resolvable by confocal light microscopy. As a consequence, the excess area stored in the shape fluctuations did not exceed a few percent of its apparent area and can therefore be ignored. If a tense vesicle with $v = 1$ is further inflated, the vesicle membrane will rupture and lose its structural integrity. This geometric constraint applies to both vesicle membranes with a uniform composition (both lipid and bound protein) and for vesicle membranes with two intramembrane domains as studied here.

However, you have raised an interesting point about preventing budding. Therefore, based on your suggestion, we tried a different approach to counteract the budding via increasing the concentration of streptavidin at the *Ld* domain.

Figure S6: Exemplary GUVs when increasing the initial concentration of Streptavidin protein (from left to right) in the lumen of the vesicles (prepared from four separate experiments) results in reduced to no membrane protrusion of the *Lo* domain after Cholera toxin addition. Scale bar corresponds to 10 μm .

In the new Figure S6 above, we have varied the final streptavidin concentration inside the vesicles while keeping the concentration of cholera toxin-B constant outside. We found that at higher concentrations, streptavidin is effective in preventing the protrusions.

The below text has been added to the main text, with the new Figure S6 in SI.

In order to establish control of synthetic cell morphology, we modulated the protein concentration in one domain whilst keeping the other one constant. To do this, we systematically increased the concentration of streptavidin (10 nM, 20 nM, 40 nM, and 80 nM) inside the vesicle while maintaining a constant CT-B Al-594 concentration outside. As shown in Supplementary Figure S6, at equilibrium (after an hour), the pronounced membrane curvature in the *Lo* domain (red) was progressively reduced as the concentration of internal bound streptavidin in the *Ld* domain (green) increased. This modulation of the protein concentration at the inner *Ld* leaflet only, counteracted the *Lo* domain curvature induced by CT-B Al-594 binding. At streptavidin concentrations of 40 nM and above, the vesicle's morphology is restored to a state close to its original spherical configuration (i.e. without CT-B).

This phenomenon mirrors the mechanism by which the outward budding of the Lo domain reduces the line free energy at the domain boundary when CT-B A1-594 is added. Therefore, we demonstrate the ability to control membrane curvature in one domain by modulating protein binding in another within a complex asymmetric phase-separated system. These findings suggest that such asymmetric systems offer a novel approach for studying the dynamic interplay between membrane domains within the same vesicle, enabling unprecedented control over domain-specific membrane curvature.

Note that we could also add more protein (CT-B) outside after the above in order to try and bring back the protrusion. However, this will be the focus of further studies, where we also modulate the protein binding more systematically and with different proteins.

We believe that these new experiments are more relevant and in-line with the ideas that have already presented in this manuscript. Moreover, controlling budded domain curvature by altering the concentration of proteins (and therefore spontaneous curvature) has not been seen in such a system previously. This further strengthens the novelty of our work. We thank the reviewer for this idea.

Regarding the stability of the protrusions. They remain stable. To demonstrate this, we have added a video (new video 7 in SI) with a vesicle having stable protrusion over a duration of >6 hours. We also want to reiterate that we don't see protrusion as a disadvantage or a lack of stability. On the contrary, we think it is a result of membrane asymmetry that is stable at equilibrium. To be able to produce these interesting shapes and control the morphology merely via asymmetric membrane construction is very interesting for us as well as to the community. In particular, for synthetic cells that become polarised and for novel drug carrier systems (i.e. anisotropic particles).

The below text has been added to reflect the above:

This pronounced curvature with less GM1 compared to previous work could be due to the enrichment of GM1 to the Lo domain. Furthermore, the protrusions remain stable for over 6 hours (Supplementary Video 7), suggests that there is no flipping of GM1 or biotinylated lipids for long time periods. This opens the possibility of using such morphologies as polarised synthetic cells or anisotropic novel drug carrier systems.

I also have some suggestions and questions regarding the experimental approach, which should be explored to better validate the new method presented in the manuscript.

3) The authors present the triple-layer inverted emulsion method as an improvement on prior work. The use of a spacer oil is interesting, and I can see how the use of oils of differing density could be used to better separate the lipid-containing squalene layer (outer leaflet) and W/O mineral oil emulsion (inner leaflet). The authors state that there is no lipid mixing during the production process, but there is no data that studies this. It would be interesting to observe whether lipid mixing occurs (perhaps through the use of fluorescent tracer lipids in the squalene and mineral oil emulsion phases), with and without the mineral oil spacer oil. This would facilitate a better understanding of the role of the mineral oil spacer, and how necessary it is in the formation of the asymmetric vesicles.

Thank you for your valuable suggestion. We appreciate your interest in the role of the mineral oil spacer in our triple-layer inverted emulsion method and the concern regarding potential lipid mixing during the production process. While we have hypothesized that the use of a lipid-free oil spacer can effectively prevent lipid mixing between the inner (mineral oil) and outer (squalene) leaflets, as we see

improvements in the asymmetry, we agree that providing direct experimental evidence would strengthen our claims.

Therefore, we performed the below experiment. With three conditions, i) double layer ii) triple layer without spacer oil, and iii) triple layer with spacer oil (from left to right). The figure below shows photographic images of inverted emulsion method performed in Eppendorf® tubes after the centrifugation step. In all conditions, PE-NBD (yellow) is added as a tracer lipid in the oil interfacing the bottom aqueous solution, while the top layered oil containing w/o droplets is with DID (blue). After performing the centrifugation, only in the condition where there is spacer oil (right), we observe is a distinct yellow colour of unmixed yellow NDB (bottom). In the condition with triple layer without spacer (centre), a blue layer can be seen above the yellow layer resulting from a partial mixing of lipids. For the double layer method (left), a homogeneous blue layer is seen resulting from significant mixing of the lipid layers. This demonstrates that there is clear mixing of lipids in the order double layer, less in the triple layer without spacer, and none in triple layer with spacer. Note, the topmost orange part (seen in all three) is a reflection of the Eppendorf rack.

This figure now added to the SI, with the following text to the main manuscript.

Figure S1: Observing the mixing of lipid dyes after the transfer of W/O droplets. The fluorophores DID (blue) and NBD-DOPE (yellow) are added to the upper and lower most oil layers respectively for (i) the double-layer (ii) triple-layer without spacer and (iii) triple-layer with spacer. The lipid dye mixing is more prominent in the case of double-layer method compared to that of triple layer method without spacer oil and with spacer oil.

To evaluate the hypothesis that the presence of spacer oil prevents lipid mixing and improves the asymmetry, we added labels to each layer. Observations are made of the mixing of the labels after the centrifugation step. Figure S1 shows that the double-layer method results in the homogeneous mixing of upper mineral oil layer with the squalene layer, and a uniform colour. In the case where the triple-layer method, but without spacer, some mixing is observed. Only in the case of the triple-layer with spacer, no oil mixing was observed. This confirms that a reduced lipid mixing in the new methodology contributed to the improved asymmetry, ~95% compared to ~80% or lower that is typically observed amongst works using the double-layer method^{32,43,44}. In addition to the density differences which maintain the separation of mineral oil and squalene, this test emphasizes the role of spacer oil in further reducing the mixing of lipids, ensuring high lipid asymmetry across the membrane.

4) In Figure 1C, the authors show fluorescence quenching data for double-layer and triple-layer emulsion phase separated-vesicles. The authors include convincing data demonstrating the asymmetry of GUVs produced via the triple-layer method, but only produce symmetric vesicles using the double-layer method. The authors cite their prior work (10.1016/j.bpj.2022.12.017) and say that the triple-layer method gave better results in this study, than the double-layer method in the previous study. However, I believe the membrane asymmetry is different, with the focus of the prior work the asymmetric incorporation of NBD-labelled bacterial lipopolysaccharides. The authors should perform fluorescence quenching on GUVs prepared via the double-layer method using the NBD-DOPE lipid used in this work, and the fluorescent quenching compared for double- and triple-layered. This, combined with 3) above will enable a better evaluation for the use of the triple-layer phase transfer method as compared to the simpler double-layer method.

We thank the reviewer for pointing out the differences. Multiple groups have already performed the double layer method with mineral oil and NBD lipids-based quenching. We assumed this data could be used to justify the differences. After recommendation by all the reviewers, we have now performed the corresponding asymmetry experiments with double layer method as well.

The plot below suggests that double layer method can achieve asymmetry albeit with large error bars ($77 \pm 6\%$) when compared to that of the results from our new triple layer method ($97 \pm 1.3\%$). A statistical analysis resulted in no significant variation between both the methods for the condition where NBD lipids are inside, but the quenching distribution for double layer method is broad compared to the triple layer method. For the condition where NBD lipids are outside, the difference is significant. Thus, solidifying our claim on high efficiency of asymmetry through triple layer method.

The below plot and text are now included in the manuscript under the “Fluorescence quenching assay to assess bilayer asymmetry” section.

Below caption for Figure 1(c) is included for clarity.,

The change in fluorescence intensity for symmetric GUVs and asymmetric GUVs induced by quenching from the addition of 10 mM sodium dithionite to the outer solution. Inserts show confocal images of exemplary GUVs before and after quenching produced via the new triple-layer method. Scale bars correspond to 5 μm . Statistical significance was determined using Mann Whitney method.

Fluorescence quenching is an established method to assess the degree of asymmetry in lipid bilayers. In our experiments, we performed a quenching assay using NBD-tagged lipids and sodium dithionite as a quencher - a commonly used combination.⁴² The quenching occurs when sodium dithionite, a reducing agent, interacts with the fluorescent NBD (7-nitrobenz-2-oxa-1,3-diazol-4-yl) group, reducing it to a non-fluorescent form. This reaction is highly selective and depends on the accessibility of the NBD fluorophore to the aqueous dithionite, providing a measure of its exposure in membrane systems.⁴³ In the first set of experiments, we compared the quenching percentage in symmetric GUVs produced from the previously reported double-layer method to our new triple-layer method. The lipid composition for this study was 99.5 mol% POPC and 0.5 mol% NBD-DOPE. As displayed in Figure 1(c), the fluorescence intensity of the symmetric GUVs produced through the double-layer method is reduced to $62 \pm 13\%$ from the unquenched membrane and a similar percentage of $53 \pm 11\%$ is observed for the GUVs produced using the triple-layer method, after the addition of sodium dithionite. This result is anticipated as the NBD-DOPE lipid gets quenched on the outer leaflet of the symmetric bilayer only,

and the intensity should reduce to half. From this data, statistically, it can be concluded that no significant difference in the percentage of quenching is observed amongst the symmetric GUVs produced from both the methods and quenching assay can be reliably performed on the asymmetric GUVs.

Two different populations of asymmetric GUVs were then produced using both methods. One comprised of GUVs produced with NBD-DOPE in the inner leaflet of the GUVs and the other had NBD-DOPE in the outer leaflet. Like the previous condition, sodium dithionite was added to the outer solution for both populations of asymmetric GUVs. In the case of NBD-DOPE in the inner leaflet, no or very few of the GUVs should be quenched for successful asymmetry. However, for the case when NBD-DOPE is in the outer leaflet (and therefore accessible to the quencher), all the fluorescence intensity should be quenched. A time-lapse video showing the quenching of asymmetric GUVs (with NBD-DOPE in the outer leaflet) in the presence of sodium dithionite can be seen in the supplementary information (Supplementary Video S1). In Figure 1(c), for the asymmetric NBD inside condition, both methods exhibit low quenching: double-layer ($14 \pm 11\%$) and triple-layer ($7 \pm 3\%$). While in the asymmetric NBD outside condition, quenching is high for both methods with $77 \pm 6\%$ for double layer method and $97 \pm 1.3\%$ for triple layer method. Interestingly, the triple-layer methodology shows lower variability in the degree of asymmetry. These results in conjunction with statistical analysis suggest that the two production methods lead to distinct quenching profiles under the tested conditions, with the triple-layer method demonstrating more consistent results (less variability) and improved asymmetry compared the double-layer method.

5) The authors report improved cholesterol integration within phase-separated GUVs due to poorer solubility of the molecule in squalene. Has this been reported previously using the double-layer method?

Indeed, the cholesterol incorporation is improved. Previous works have not attempted to produce phase separation with squalene. To better explain this, we have now added a new table to the SI (Table S4) where previous attempts were made with mineral oil alone (double-layer method) to achieve phase separated GUVs and have shown contrasting results. Moreover, our own experiments using the double layer method (mineral oil alone), resulted in lower levels of cholesterol incorporation leading to $<1\%$ of the GUVs showing liquid-liquid phase separation. We have now included a new Figure S3 below.

Table S4: Various phase transfer methods using mineral oil and $\geq 50\%$ cholesterol to induce membrane phase separation in lipid-based vesicles.

Method	Lipid composition	Oil used	% GUVs with liquid-liquid phase separation	Ref.
cDICE	1/1 DOPC/DPPC and 100% cholesterol	Mineral oil	$<1\%$	Blosser et. al. 2016
Inverted emulsion	1/1 DOPC/DPPC and 40 to 80% cholesterol or 1/1 DOPC/EggSPM and 40 to 80% cholesterol	Mineral oil	95%	Karamdad et. al. 2018
Inverted emulsion	1/1 DOPC/DPPC and 50% cholesterol or 1/1 DOPC/EggSPM and 50% cholesterol	Mineral oil	$<1\%$	This study

Triple layer inverted emulsion	1/1 DOPC/EggSPM and 30% to 60% cholesterol	Mineral oil & Squalene	>60%	This study
--	------------------------	------	------------

In the below Figure we used the double layer method, and we were unable to reproduce the work performed by Karamdad et al. (2018) for both DPPC and Sphingomyelin conditions.

Figure S3: Symmetric GUVs with phase-separation compositions (50% Chol, DOPC:SPM – 1:1 & 50% Chol, DOPC:SPM – 1:1) produced using the double-layer method and showing no phase separation. Scale bar corresponds to 10 μm .

Hence, we developed this new method to overcome this issue. It is clear from the literature that varying results exist. Hence, we aimed (and succeeded) to present a new method that we hope the community will use. Or at least provide more method options.

Moreover, compared to the findings of Karamdad et al. (2018), our triple-layer method achieves results that are well-aligned with those obtained using the electroformation (i.e. no oil) technique.

Below text is added to the main:

While electroformation based production of phase separated GUVs is a common strategy, it is challenging to produce them using inverted emulsion method. Previous studies demonstrated very poor incorporation of cholesterol when mineral oil was used.^{46,47} except for one study⁴⁸ (see Table S4). However, we are unable to reproduce the phase separation using double layer method involving mineral oil alone (Supplementary Figure S3). This inconsistency motivated us to develop a new method.

Together with this data, we can confirm that cholesterol incorporation is better using our new triple layer method with squalene.

Is the cholesterol present in the squalene layer of the assembled phase column only, or also in the mineral oil emulsion that is layered on top?

Cholesterol is present in both of the layers (but not the spacer). We have added the following text to the manuscript to make this point clearer.

GUVs were formed with varying concentrations of cholesterol (in both the squalene and mineral oil phases) while keeping the sphingomyelin to DOPC ratio constant (i.e., 1:1).

Note that lipid compositional information (per leaflet) is also provided in Tables S2 and S3, but now we have amended the tables to explicitly say which oils phases they are in.

6) Could the authors provide more data for the curvature of the Lo phase for i) asymmetric phase separated vesicles and ii) the symmetric phase separated vesicles after Ct-B AL 594 binding? In each case only one GUV is presented (asymmetric Fig 4C, symmetric Fig S3). For Fig S3, in the overlay of the image the domains do not align (probably due to timelapse between taking measurements in different fluorescence channels). Providing images of more GUVs would enable the reader to better evaluate the range of curvatures shown for asymmetric/symmetric phase separated GUVs respectively.

Thank you for noticing the issue. Figure S5 is now updated to better represent the lack of curvature compared to the GUVs with asymmetric phase separation. More images are uploaded in the supplementary material now under Figure S8.

Figure S5: Symmetric GUVs with domain phase-separation (50% Chol, DOPC:SPM – 1:1 1% GM1 in the 0.5% DiD) showing no protrusions. CT-B binding to the liquid-ordered region (red channel) and DiD to the liquid disordered region (green channel). Scale bar corresponds to 10 μm .

7) The shape analysis of vesicle curvature is interesting but is only applied to a small number of asymmetric GUVs (for curvature discontinuity, 6 GUVs are evaluated and for the closed membrane necks 10 GUVs are evaluated). Were these GUVs prepared from a single-phase transfer experiment? It would be good to apply this method to a larger number of GUVs, to better understand the range of curvatures observed the frequency with which the protrusion of the Lo domain occurred across a vesicle population.

We have now analysed and added 4 more GUVs for the curvature discontinuity data to strengthen findings. These analyses are taken from N = between 3 or 4 number of different phase transfer preparations. We all did not see a large variability in the resulting analysis for each condition which also strengthens the data (and method). We have now updated Table S5 (below) to reflect this, as well as including additional figures to Figure S8 (below). We thank the reviewer for this and hope this is now

sufficient. Similar extensive analysis (16 GUVs, previously 10) also performed with vesicles undergoing neck closure – associated data is presented in Table S6 and a new Figure S9.

Table S5: Membrane curvature discontinuity analysis across the domain boundary of two-domain asymmetric GUVs.

GUVs	Ro, se mi	Rd, se mi	R _d _b	ψ	C _{1o}	C _{1d}	C2 = sin(ψ)/r	M _o	M _d	M	R _{npol}	R _s _{pol}	C2= 1/R _{spol}	M _o _{db}	M _d _{db}	M
1	4.98	0.52	4	72	-0.2008	1.923077	0.237714865	0.018456	1.080396	1.098852	3.95	8.1	0.12345679	-0.03867	1.023267	0.984594
2	4.2	0.32	3.415	60	-0.2381	3.125	0.25351679	0.007711	1.689258	1.696969	3.12	6.56	0.152439024	-0.04283	1.63872	1.595891
3	61.1	9.45	5.33	40.5	-0.01637	0.10582	0.121796532	0.052715	0.113808	0.166523	5.52	10.19	0.098135427	0.040884	0.101978	0.142862
4	17.48	16.61	6	59	-0.05721	0.060205	0.142816386	0.042804	0.101511	0.144315	5.27	8.67	0.115340254	0.029066	0.087772	0.116838
5	4.33	1.69	4.73	40	-0.23095	0.591716	0.135838573	-0.04755	0.363777	0.316223	3.32	13.18	0.075872534	-0.07754	0.333794	0.256257
6	4.44	1.37	2.44	33	-0.22523	0.729927	0.223112349	-0.00106	0.47652	0.475463	3.38	6.79	0.147275405	-0.03897	0.438601	0.399626
7	8.71	0.48	6.68	70	-0.11481	2.083	0.1406	0.00129	1.1119	1.1249	6.14	10.3	0.09708	-0.00886	1.09021	1.081349
8	5.55	0.09	2.63	83	-0.18018	11.111	0.37735	0.09859	5.744235	5.8428	3.21	7.26	0.137741	0.002122	5.624426	5.603207
9	5.01	2.4	3.16	51	-0.1996	0.41667	0.2458423	0.023121	0.331255	0.354375	4.98	6.85	0.1459854	-0.002681	0.281326	0.254518
10	2.45	1.98	2.41	49	-0.40816	0.505051	0.313039	-0.04756	0.409045	0.361483	3.51	5.52	0.18115942	-0.01135	0.343105	0.229603

Figure S8: Confocal images of six GUVs along with the analysis of their shape contours as in Figure 5, with annotations using Autodesk AutoCAD®. Scale bar corresponds to 10 μm .

Figure S9: Confocal images of 16 GUVs with neck closure, fitted with a spherical circle for both green segment (Ld phase) and red segment (Lo phase), with annotations using Autodesk AutoCAD®. Scale bar corresponds to 10 μm.

Table S6: Effective Membrane curvature of two domain asymmetric GUVs with neck closure.

No.	R _i	R _j	1/M _i	1/M _j	M _{ij}
-----	----------------	----------------	------------------	------------------	-----------------

1	3.45	6.5	0.289855	0.153846	0.221851
2	4.34	4.48	0.230415	0.223214	0.226815
3	3.1	11.4	0.322581	0.087719	0.20515
4	4.66	7.8	0.214592	0.128205	0.171399
5	7.45	28.48	0.134228	0.035112	0.08467
6	3	19.82	0.333333	0.050454	0.191894
7	1.4	4.15	0.714286	0.240964	0.477625
8	2.1	8.4	0.47619	0.119048	0.297619
9	2.8	6.35	0.357143	0.15748	0.257312
10	1.85	6.6	0.540541	0.151515	0.346028
11	1.57	5.4	0.636943	0.185185	0.411064
12	1.5	6.12	0.666667	0.163399	0.415033
13	2.4	4.65	0.416667	0.215054	0.31586
14	2.7	16.25	0.37037	0.061538	0.215954
15	2.3	8.01	0.434783	0.124844	0.279813
16	1.89	6.11	0.529101	0.163666	0.346383

Finally, the methods section needs to be improved to enable reproducibility of the work;

We thank the reviewer for pointing out these discrepancies.

8) For osmolarity-induced bud division experiments, the final solution osmolarity is not described (only that 2.5 ul of 2M glucose solution is added). Please amend this so it is clear to the reader the final external solution osmolarity (and the osmotic gradient across the membrane).

Methods section is updated, and the new information is added for clarity and reproducibility.

For osmolarity induced bud division experiments, 2.5 μ L of 2 M glucose solution was injected into the outer solution creating a 50 mOsm hypertonic solution to deflate the GUVs, after cholera toxin binding.

9) The description of shape analysis in the method is minimally described - the only detail is that Autodesk AutoCAD was used to fit the vesicles. Please give sufficient information for a reader to reproduce the shape analysis.

A new methods section describing the shape analysis and fitting is added. In addition, an exemplary CAD file with a fitted vesicle is now included with the manuscript as a supplementary file.

Vesicle shape contour analysis

For shape analysis, Autodesk® AutoCAD® was used to fit the vesicles. An exemplary CAD file for the GUV fitting method is included in the supplementary information. To perform the contour analysis, the GUV image file with scale bar is imported into the AutoCAD® file and the image is adjusted to match the dimensions of the GUVs to that of the CAD file via scaling (in the example file, the scale bar is 10 μm). In the next step to draw axi-symmetric line passing along the GUVs centre (Figure 5(a)), two osculating circles fitting the Ld (green) and Lo (red) domains are drawn with radii, R_{spol} and R_{npol} , respectively (Figure 5(b)). Next, connecting the centres of the two circles, a line is drawn, representing the symmetrical axis. This is followed by drawing a line perpendicular to the axi-symmetric line to the domain boundary (where Ld domain (green) meets the Lo domain (red)), shown as a dashed line in Figure 5(b). A semicircle is drawn with its corner at the domain boundary (left or right side of the GUV, in Figure 5(b), it is drawn on the right side) having a radius of $R_{\text{o,semi}}$, until the arc of the semicircle fits the Lo domain (red) as shown in Figure 5(b) and Figure S8. A line is drawn along and parallel to the diameter of the semicircle until it meets the axi-symmetric line giving an angle, tilt angle (ψ) (Figure 5(b)). Thus, obtained measurements are used to calculate various parameters using the equations mentioned in the main text as well as in SI and presented in Table S5.

Similarly, the effective mean curvatures (M_{ij}^{eff}) for vesicles with closed neck configurations are calculated by measuring the radii of the green region (R_i) and red region (R_j). The radii were measured by fitting a circle using circle function in AutoCAD® (an example fit can be found in the CAD file uploaded as in the supplementary information).

10) How are the mathematical calculations performed? Please give details.

Mathematical calculations are presented as a table in the SI, along with all the relevant equations. Another reviewer has requested the source data which we will upload to the SI as well.

The following has been added to make this location clearer in the main text.

The corresponding data obtained after fitting the vesicles (refer to Methods section, Vesicle shape contour analysis and Supplementary CAD file) are provided in Table S5 of the Supplementary Information.

Reviewer #2 (Remarks to the Author):

This is a very interesting manuscript. It's well written, details are clear, data presentation is very logical, and conclusions are generally well justified. The curvature formalism capable of considering the lipid mixture and effects of lipid ordered and lipid disordered phase is appreciably rigorous, and I haven't seen that applied in the field of synthetic cell engineering before. Authors used more biologically relevant lipids, which increases the value of this work. Authors also interestingly claim better incorporation of cholesterol.

Addressing few issues would improve this manuscript:

The authors mention that others have proffered methods for preparing asymmetric membranes, and the manuscript compares double layer emulsion to triple layer emulsion in figure 1. However, the authors did not compare the ability to form asymmetric bilayers between their method (triple layer emulsion) and double layer emulsion.

Moreover, authors significantly modify the lipid compositions and emulsion formulation between published double layer emulsions and what they test as double layer here. That they didn't overtly compare asymmetry between the double and triple layer methods is odd too because they validate that their assay would work with both methods.

We thank the reviewer for pointing out this discrepancy. We understand that our new method had significant updates so a comparison to previous methods was not correct. As a result, we have now performed additional experiments where we also perform the previous double later method (keeping lipids and oils the same as our new methods) and included them in the manuscript.

The graph below suggests that double layer method can achieve asymmetry albeit with large error bars when compared to that of the results from triple layer method. A statistical analysis resulted in no significant variation between the two methods for the condition where NBD lipids are inside, but the quenching distribution for double layer method is broad ($77 \pm 6\%$) compared to the triple layer method ($97 \pm 1.3\%$). In addition to this, for the condition where NBD lipids are outside, there is a significant difference in the data. Thus, solidifying our claim on high efficiency of asymmetry through triple layer method.

The below figure has been added to the main text under the "Fluorescence quenching assay to assess bilayer asymmetry" section. Note that reviewer #1 (comment no. 4) also pointed out. Please see that response for main text changes.

Below caption for Figure 1(c) is included for clarity.,

The change in fluorescence intensity for symmetric GUVs and asymmetric GUVs induced by quenching from the addition of 10 mM sodium dithionate to the outer solution. Inserts show confocal images of exemplary GUVs before and after quenching produced via the new triple-layer method. Scale bars correspond to 5 μm . Statistical significance was determined using Mann Whitney method.

Authors were not adequately specific about how successful they were with cholesterol incorporation beyond saying that they were able to see changes in phase separation dynamics with a range of working concentrations during the preparation.

It is worth pointing out that previous attempts from the literature show that the incorporation of cholesterol within vesicles produced using the invited emulsion (i.e. phase transfer) method do not result in phase separation, except for one published instance. It was this low number of reports and the inconstancy which motivated this work. A summary of these attempts (including our new data) is now shown in the new table below (Table S4 included in the SI). Our own experiments using double layer (mineral oil, alone), resulted in lower levels of cholesterol incorporation leading to <1% of the GUVs showing liquid-liquid phase separation (see figure new S3 below, added to SI). However, after the implementation of our new triple-layer method, we can confirm that more than 60% of the GUVs are phase separated (note: 50% or more is considered a phase separated batch in the community). Together with previous work, we can confidently say that cholesterol incorporation is indeed improved, from <1% of the population to >60% of the population having liquid-liquid phase separation. Moreover, we also compared our new results to the “gold standard” for phase separation,

i.e. electroformation (a none oil method), and achieved very close data. We hope that the new SI Table and Figure (with a direct comparison) makes it clear how much of an improvement our new method is for creating reproducible phase separated membranes.

Table S4: Various phase transfer methods using mineral oil and $\geq 50\%$ cholesterol to induce phase separation in lipid-based vesicles.

Method	Lipid composition	Oil used	% GUVs with liquid-liquid phase separation	Ref.
cDICE	1/1 DOPC/DPPC and 100% cholesterol	Mineral oil	<1%	Blosser et. al. 2016
Inverted emulsion	1/1 DOPC/DPPC and 40 to 80% cholesterol or 1/1 DOPC/EggSPM and 40 to 80% cholesterol	Mineral oil	95%	Karamdad et. al. 2018
Inverted emulsion	1/1 DOPC/DPPC and 50% cholesterol or 1/1 DOPC/EggSPM and 50% cholesterol	Mineral oil	<1%	This study

PC:DPPC 1:1 – Chol 50%

PC:SM 1:1 – Chol 50%

Figure S3: Symmetric GUVs with phase-separation compositions (50% Chol, DOPC:SPM – 1:1 & 50% Chol, DOPC:SPM – 1:1) produced using the double-layer method and showing no phase separation. Scale bar corresponds to 10 μm .

The following is added to the main text:

While electroformation based production of phase separated GUVs is a common strategy, it is challenging to produce them using inverted emulsion method. Previous studies demonstrated very poor incorporation of cholesterol when mineral oil was used.^{46,47} except for one study⁴⁸ (see Table S4). However, we are unable to reproduce the phase separation using double layer method involving mineral oil alone (Supplementary Figure S3). This inconsistency motivated us to develop a new method.

Interesting methods: The spacer oil that separates the two lipid-in-oil solutions that contain the intended inner and outer leaflet lipid compositions,

Strictly speaking Kruskal-Wallis should never be conflated with ANOVA, also I want more explanation as to why the authors used something really exotic here. They chose a nonparametric multi category test, if nonparametric due to dynamics of liposome formation, why not just Mann-Whitney with Bonferroni correction. There should be justification for nonparametric tests (even though they are always more stringent than parametric tests) and explanation for the choice of Kruskal Wallis over the more common Mann-Whitney.

With the addition of new data, we found that it is indeed relevant to use Mann-Whitney and compare each category between double and triple layer methods. Before, we didn't have the double layer asymmetric data. After the addition of this, we used more common Mann-Whitney. Now written in caption (see below). We thank the reviewer for suggesting this analysis method.

Statistical significance was determined using Mann Whitney method.

The authors say that cholesterol is more soluble in mineral oil than in squalene, which I am inclined to believe, but the manuscript don't give citation or explanation as to why (the explanation to me would be that mineral oil has cyclic hydrocarbons and aromatics). Likewise there are many small molecule quenchers used, I would like a citation or quick mechanistic comment about the sodium dithionate.

Indeed this information was missing. For the cholesterol: the entire system also involves partitioning to either mineral oil, squalene, aqueous solution and/or the lipid monolayer. This would make measuring the partitioning of cholesterol in a complex system challenging due to the interplay of multiple factors. However, we have now added the following text to the discussion section:

This is due to the fact that mineral oil contains more hydrocarbon chains that are saturated compared to squalene. Cholesterol is known to have a lower solubility in unsaturated oils which results in a higher partitioning towards the lipids formed at the oil-water interfaces with squalene.⁶⁶

Regarding small molecules quencher used in this study, this the following explanation has been included in the main text, together with a reference.

The quenching occurs when sodium dithionite, a reducing agent, interacts with the fluorescent NBD (7-nitrobenz-2-oxa-1,3-diazol-4-yl) group, reducing it to a non-fluorescent form. This reaction is highly selective and depends on the accessibility of the NBD fluorophore to the aqueous dithionite, providing a measure of its exposure in membrane systems.⁴³

Overall, it's a very interesting paper. I believe after fixing and clarifying the above-described issues, it will be a valuable publication of interest to our community.

The authors would like to thank the reviewer for this comment.

Reviewer #3 (Remarks to the Author):

The manuscript by Yandrapalli et al. describes the development of an emulsion phase transfer method to generate giant unilamellar vesicles with an asymmetric lipid bilayer. This is an interesting concept, but I have a number of concerns that need to be address, especially regarding the controls used to quantify if the method is an improvement over existing procedures.

1) Page 3. In the introduction, the authors state that “In contrast, the outer leaflet is devoid of such negatively charged lipids”. While they are correct PS is not present in the outer membrane of healthy cells, the outer membrane is still negatively charged using alternative lipids. If this were not the case, drug delivery with positively charged polymers/lipids would not work.

Thank you for your observation. You are correct that, while phosphatidylserine (PS) is typically absent from the outer leaflet of healthy cell membranes, the outer leaflet can still carry a net negative charge due to the presence of alternative negatively charged lipids, such as gangliosides or other sialylated glycolipids. This is a critical factor in the interaction of positively charged polymers or lipids with cell membranes for drug delivery purposes. We will revise this statement in the manuscript to clarify the nuanced distribution of negatively charged lipids in the outer membrane and avoid the implication that it is entirely neutral or lacking negative charge.

We will update this section to reflect the presence of these alternative negatively charged lipids, ensuring accurate representation of membrane charge characteristics. The following text has been changed:

While phosphatidylserine (PS) is indeed absent from the outer leaflet of healthy cell membranes, the outer leaflet can still possess a net negative charge due to the presence of alternative negatively charged components, such as gangliosides or sialylated glycolipids. These lipids contribute to the overall negative surface potential of the outer membrane, which is critical for biological processes such as cell-cell interactions and the efficacy of drug delivery systems utilizing positively charged polymers or lipids.¹⁴

2) Page 4. The authors have not included droplet interface bilayers as other methods to create asymmetric bilayers.

Thank you for pointing this out. We will update the manuscript to include DIBs as a complementary technique, highlighting their utility in studying membrane biophysics and protein-lipid interactions.

The following has been added to the main text:

Methods such as droplet interface bilayers (DIBs)¹⁷, vesicle (hemi) fusion to a supported lipid bilayer (SLB)¹⁸, or the use of methyl- β -cyclodextrin¹⁹ offer ways to produce asymmetric bilayers but the resulting membrane structures are limited. In the case of DIBs, the presence of large amounts of oil can be an issue for specific proteins or lipids, and subsequent manipulation (e.g. to add biomolecules from outside) after the formation of the bilayers is not possible.

3) Page 4/5. There needs to be more discussion of previous methods to produce asymmetric bilayers and their drawbacks/differences to what the authors have developed. For instance, the sentence “Presently, to our knowledge, there is no methodology that can produce asymmetric membranes, along with the complexity of phase separation that can also sustain protein functionality.” needs to be more thoroughly evidences and explained.

We agree and have now extended this discussion in the introduction to better compare alternative methods and to explain why we developed this method and claim that no other method has been shown

to make phase separated asymmetric membranes. This now included methods such as DIBs as well as drawbacks/differences as suggested.

Methods such as droplet interface bilayers (DIBs)¹⁷, vesicle (hemi) fusion to a supported lipid bilayer (SLB)¹⁸, or the use of methyl- β -cyclodextrin¹⁹ offer ways to produce asymmetric bilayers but the resulting membrane structures are limited. In the case of DIBs, the presence of large amounts of oil can be an issue for specific proteins or lipids, and subsequent manipulation (e.g. to add biomolecules from outside) after the formation of the bilayers is not possible. In the case of vesicle fusion-based asymmetry generation, usage of divalent cations/oppositely charged lipids could add to the complexity. Moreover, the resulting SLB/GUV has an inaccessible inner leaflet or requires lipid flip-flop to alter its composition. The methyl- β -cyclodextrin can produce asymmetric membranes but is limited to specific lipid types with limited access to the vesicle lumen and is process intensive. For these reasons, cell-sized free-standing GUVs with accessible outer and inner leaflets is desired. The latter point is an important one as not all GUV production methods provide high encapsulation. Being able to visualize them and encapsulate larger biomolecules within makes them a strong candidate for building synthetic cells from the bottom-up²⁰. Techniques like electroformation and hydration methods only yield symmetric bilayers and are further limited by difficulties with using physiological buffers or low encapsulation efficiencies. Encapsulation is essential when modelling cells, therefore researchers require not only asymmetric membranes but also high encapsulation of various biomolecules.²¹ More sophisticated techniques such as microfluidics have been shown to create GUVs with a variety of lipid asymmetries with excellent encapsulation efficiencies.²²⁻²⁵ For example, pulsed-jetting technique was used to create cell-size lipid vesicles with asymmetric lipid bilayers²⁶, and double emulsion-based templating is adopted to create lipid vesicles with biotinylated lipid asymmetry²⁷ as well as bacteria-like vesicles with lipopolysaccharides in the outer leaflet²⁸.

4) Page 7. The authors state “To achieve a high degree of bilayer asymmetry in GUVs, the challenge is to prevent mixing of the two different lipid containing oil solutions. Failure to do so will result in contamination of inner leaflet and outer leaflets with the opposite lipids and decrease the degree of asymmetry.” This is not evidenced. Furthermore, these are the critical control experiments that are not performed throughout the paper. The control vesicles are prepared symmetrically, but this is not what the authors are claiming they are improving on. There needs to be controls for alternative state of the art asymmetric vesicle preps, for instance those generated by the authors themselves (doi: 10.1016/j.bpj.2022.12.017) and/or just adding an emulsion formed with the inner lipid to a phase transfer column prepared with the outer lipid. These controls are required throughout the work (especially Fig 1, 2, and 4) to actually back up the claims that they are improving on previous methods.

Thank you for your valuable suggestion. We appreciate your interest in the role of the mineral oil spacer in our triple-layer inverted emulsion method and the concern regarding potential lipid mixing during the production process. While we have hypothesized that the use of a lipid-free oil spacer can effectively prevent lipid mixing between the inner (mineral oil) and outer (squalene) leaflets, as we see improvements in the asymmetry, we agree that providing direct experimental evidence would strengthen our claims.

Therefore, we performed the below experiment. With three conditions, i) double layer ii) triple layer without spacer oil, and iii) triple layer with spacer oil (from left to right). The figure below shows photographic images of inverted emulsion method performed in Eppendorf® tubes after the centrifugation step. In all conditions, PE-NBD (yellow) is added as a tracer lipid in the oil interfacing the bottom aqueous solution, while the top layered oil containing w/o droplets is with DID (blue). After performing the centrifugation, only in the condition where there is spacer oil (right), we observe is a distinct yellow colour of unmixed yellow NDB (bottom). In the condition with triple layer without spacer (centre), a blue layer can be seen above the yellow layer resulting from a partial mixing of lipids.

For the double layer method (left), a homogeneous blue layer is seen resulting from significant mixing of the lipid layers. This demonstrates that there is clear mixing of lipids in the order double layer, less in the triple layer without spacer, and none in triple layer with spacer. Note, the topmost orange part (seen in all three) is a reflection of the Eppendorf rack.

This figure now added to the SI, with the following text to the main manuscript.

Figure S1: Observing the mixing of lipid dyes after the transfer of W/O droplets. The fluorophores DID (blue) and NBD-DOPE (yellow) are added to the upper and lower most oil layers respectively for (i) the double-layer (ii) triple-layer without spacer and (iii) triple-layer with spacer. The lipid dye mixing is more prominent in the case of double-layer method compared to that of triple layer method without spacer oil and with spacer oil.

To evaluate the hypothesis that the presence of spacer oil prevents lipid mixing and improves the asymmetry, we added labels to each layer. Observations are made of the mixing of the labels after the centrifugation step. Figure S1 shows that the double-layer method results in the homogeneous mixing of upper mineral oil layer with the squalene layer, and a uniform colour. In the case where the triple-layer method, but without spacer, some mixing is observed. Only in the case of the triple-layer with spacer, no oil mixing was observed. This confirms that a reduced lipid mixing in the new methodology contributed to the improved asymmetry, ~95% compared to ~80% or lower that is typically observed amongst works using the double-layer method^{32,43,44}. In addition to the density differences which maintain the separation of mineral oil and squalene, this test emphasizes the role of spacer oil in further reducing the mixing of lipids, ensuring high lipid asymmetry across the membrane.

Below is the new Figure added under the “Fluorescence quenching assay to assess bilayer asymmetry” section. Please see Reviewer #1’s comment for additional text as they also asked for this new experiment.

Below caption for Figure 1(c) is included for clarity.,

The change in fluorescence intensity for symmetric GUVs and asymmetric GUVs induced by quenching from the addition of 10 mM sodium dithionite to the outer solution. Inserts show confocal images of exemplary GUVs before and after quenching produced via the new triple-layer method. Scale bars correspond to 5 μm . Statistical significance was determined using Mann Whitney method.

5) Page 9. Authors state “To avoid this, either the surface area of the interface between squalene and glucose solution has to be increased or the number of W/O droplets needs to be reduced.” Can the lipid concentration be increased?

Only to an extent, we have studied the role of lipid concentration on the vesicle formation in our previous work – Moga et. al. 2019. Based on this, we are at the maximum concentration of lipids. We believe that the rate of transfer of droplets across the monolayer is greater than the rate of formation of lipid monolayer at the oil-water interface. i.e. the monolayer needs 30 or more minutes to form, whereas centrifugation is 3 min.

We have updated the main text to reflect this.

However, increasing the concentration of the lipids does improve monolayer formation in general, but beyond 200 μM we do not see a significant improvement.³³ Considering, the time implemented to form the lipid monolayer is 30 min and the centrifugation time is 3 min, the rate of formation of lipid monolayer is much longer than the rate of droplet transfer.

6) Page 10. Authors state “The lipid composition for this study was 99.5 mol% POPC and 0.5 mol% NBD-DOPE.” Did the asymmetric quantification change when different % of the asymmetric lipid was used. I.e. only 0.5% is put on one side. This is the case in later experiments too, when only a small % is made asymmetric.

In earlier studies it has been shown with double layer method that doubling the concentration of asymmetric lipid from 5% to 10% resulted in similar quenching percentages. (<https://doi.org/10.1021/acs.langmuir.3c03370>) Thus, we don't expect any difference in quenching percentage despite increasing the concentration of lipid to a certain degree. However, we can speculate that an increase in the order of 10x or more might result in interesting observations. We believe at this stage it is not in the scope of this paper. Another interesting point might be to increase the % of asymmetric lipids for the protein binding experiments, to test the limits. This will be the focus of future work.

We have added the following text for clarification.

Bilayer asymmetry is proven via quenching experiments, first with simple lipid mixtures containing 0.5% of PE-NBD as the asymmetric lipid and later with complex lipid mixtures containing up to 1% of large head group lipids as asymmetric lipid. Even with experimental observations lasting well beyond 6 h, the observed asymmetry and the resulting membrane curvature remained stable for the entire duration. Although we haven't tested higher concentrations of lipid asymmetry, up to 10 mol% in asymmetric lipid concentration within vesicles showed similar percentage of asymmetry in previous work.⁴³ However, it would be worthwhile to study asymmetric lipid concentrations beyond 10% to achieve higher relevance to eukaryotic membranes.

7) Page 11. Authors state “This suggests that a significant amount of the GUVs, 93 %, remained asymmetric for the duration of the experiment (approximately 1 hour) with no observable permeation or morphological changes.” Did the authors measure past 1 hour? This seems like a very short time to quantify.

This is a good point. Yes, experiments involving protein asymmetry are acquired well beyond half a day with no change or appearance in protein binding to flipped lipids, if any.

The following text has been added.

Even with experimental observations lasting well beyond 6 h, the observed asymmetry and the resulting membrane curvature remained stable for the entire duration.

8) Fig 2. Why did the authors only test the single asymmetric membrane by adding the proteins from the outside? Looking at 2e/f the streptavidin signal is much weaker at the membrane this way (albeit with the other lipid on the outside too), is there a reason for this?

Indeed, for Figure 2 b (i.e. single protein experiment) we added streptavidin externally. The main reason was that if we encapsulated it, there would be limited streptavidin and less signal. So, we chose to show the single protein binding experiment by adding it externally. i.e. for best visibility.

9) Page 20. Authors state “Both streptavidin (inside) and CT-B AL-594 (outside) proteins were used to further enhance the asymmetry and applicability of the system.” Please explain this in more detail.

We appreciate the reviewer's suggestion; this could have been clearer. We will provide additional clarification. Rather than simply demonstrating asymmetry via NBD-based quenching, as done in many

studies, by incorporating streptavidin inside and CT-B AL-594 outside, we 1) prove again that the membrane is asymmetric and 2) enhance both the functional asymmetry and applicability of the system for future studies involving asymmetric binding of various proteins.

The following text is now included in the main text for clarification.

Such a design mimics the natural asymmetry present within biological membranes (distinct lipid types and peripheral proteins on either side of the membrane), creates distinct internal and external functionalities, and broadens the system's use for modelling of biomembranes. For example, this setup could be used to study the selective binding of signalling proteins to one side of the membrane while simultaneously investigating receptor-ligand interactions or membrane curvature on the external membrane surface. In general, we believe that more complex asymmetric membranes will be crucial for increasing the functionality of synthetic cells, either within the field of bottom-up synthetic biology or for applications in biotechnology and biomedicine.⁴⁹

10) Page 20 and throughout. There needs to be a better way of explaining the lipid mix and which are in the outer/inner. For instance, "DOPC:SPM:Chol - 1:1:2, 0.2 mol% biotinyl cap PE, 1 mol% GM1 and 0.5 mol% DiD" does not explain the system well enough.

We have added the following format for clarification. DOPC:SPM:Chol - 1:1:2, with 0.2 mol% biotinyl cap PE in the inner leaflet, and 1 mol% GM1 in the outer leaflet. In addition, Table S1 and S2 in the SI also mention the leaflet composition of every condition used in this study.

11) Page 22. Authors state "In contrast, when adding CT-B AL-594 to symmetric phase-separated GUVs, no such protrusion was observed (Figure S3)." Can they explain why this might be? I.e. the presence of GM1 in the inner phase stops the protein from protruding the phase, but this is not obvious. Also on the same page there is another sentence that is not explained and difficult to understand "Note that there is no binding of streptavidin in the Lo domain (red) suggesting that there are no biotinylated lipids in the Lo phase, and that there is no large head group bearing lipids on the inner leaflet to compensate for the asymmetry of the GM1 in the outer leaflet unlike in symmetric phase-separated GUVs".

In the case of symmetric phase separated GUVs, there are large head group lipids such as GM1 on both sides of the leaflet, and CT-B on the outside. In the asymmetric case, the only difference is the lack of GM1 on the inner leaflet of the Lo domains. Thus, the asymmetry is pronounced (lipid asymmetry + protein asymmetry). This lack of pronounced asymmetry in symmetric phase-separated GUVs prevented them from undergoing such membrane protrusions.

The following has been added:

Thus, the outward-budding of the Lo domain is driven by two synergetic mechanisms. First, the GM1-enriched outer leaflet leads to a positive spontaneous curvature of this domain, arising from the large head groups of the GM1 glycolipids. Second, the domain boundary between the Lo- and the Ld-domain has a positive line tension. Therefore, the free energy of the Lo-domain is reduced when the budding decreases the length of the domain boundary.⁵⁰

The sentence – "Note that there is no binding of streptavidin in the Lo domain (red) suggesting that there are no biotinylated lipids in the Lo phase, and that there is no large head group bearing lipids on the inner leaflet to compensate for the asymmetry of the GM1 in the outer leaflet unlike in symmetric phase-separated GUVs" is now moved to earlier part of the section, Production of asymmetric phase-separated membranes, for more clarity. Below is the modified sentence added to the main text.

This is confirmed through the binding of streptavidin and CT-B A1-594 proteins that have high specificity towards biotin and ganglioside (Figure 4(a) and Supplementary Figure S5).

12) With regards to the curvatures quantified. There is mathematical modelling of individual vesicles, but no discussion or analysis of how they compare or how consistent they are. Or how these values change with altered composition/concentrations.

We appreciate the reviewer's valuable comment. We recognize the need to explicitly synthesize and summarize this comparative data in the main text. In response, the manuscript now provides detailed quantitative comparisons of vesicle curvatures across multiple individual vesicles and conditions. Below is our response structured into three sections addressing: (1) membrane discontinuity analysis and (2) neck closure analysis consistency across vesicles as well as discussing the complexity. We have calculated the average differences ($0.87 \pm 0.60 \mu\text{m}^{-1}$) amongst the discontinuity observed across all the vesicles and vesicles with neck closure having a mean effective curvature of $0.307 \pm 0.034 \mu\text{m}^{-1}$ is observed.

To assess the consistency and tunability of membrane curvatures, we analysed a set of ten GUVs using shape contour fitting (Table S5) and sixteen vesicles undergoing neck closure (Table S6). Across all GUVs, curvature discontinuities at the domain boundary were consistently observed. Our shape analysis further revealed a significant discontinuity in mean curvature across the domain boundary, quantitatively expressed as an average difference of $0.87 \pm 0.60 \mu\text{m}^{-1}$ (Table S5). This variability underscores the influence of local geometric factors (varying radii of L_o and L_d segments within the population) on membrane curvature.

In addition, we have added the follow text to discussion to better summarise the results

Analysis of vesicles undergoing budding and division showed a mean effective neck curvature of $0.307 \pm 0.034 \mu\text{m}^{-1}$; Table S6), demonstrating a consistent curvature within the population. These results demonstrate the reproducibility and controllability of curvature generation in our system. The shape analysis conducted is based on the theory of curvature elasticity as reviewed⁶⁵ and extended in⁶⁷. The basic ingredients of this theory are described in the SI, Equations S1 - S8. In addition, this theory is applied to curvature discontinuity at domain boundary and to closed membrane neck configurations. Inspection of these equations shows that the vesicle shapes depend, in general, on the bending rigidity κ_o and the spontaneous curvature m_o of the L_o domain, on the corresponding curvature-elastic parameters κ_d and m_d of the L_d domain, as well as on the line tension λ of the domain boundary. To obtain a quantitative comparison between theory and experiments, one would have to measure these five membrane-elastic parameters, which is beyond the scope of the present manuscript. On the other hand, our analysis demonstrates that the mean curvature of the membrane is discontinuous across the domain boundary, a generic morphological feature that has not been addressed in previous studies. Furthermore, our analysis also provides a lower bound of 20 pN for the constriction force, f , acting against the closed membrane neck.

13) Page 29, authors state "These unique synthetic cells with complex membranes were able to exhibit cell-like behavior, with budding and division events, from a purely biophysical origin with no additional energy input." Is this fair if the authors are adding protein and putting the vesicles under osmotic pressure to attain these features?

Our phrase "with no additional energy input" referred to energy input arising from the coupling of membrane fission to chemical reactions such as ATP or GTP hydrolysis. In order to clarify this point, we have now rephrased the text on page 29 which now reads: "...from a purely biophysical origin with no chemomechanical coupling of membrane fission to chemical reactions such as ATP or GTP hydrolysis". Thanks for pointing this out.

14) Methods. The authors use the phrase 'briefly' multiple times. The materials and methods should be exhaustive for someone to repeat. Regarding density of squalene vs mineral oil, this depends on the mineral you purchase, which is not given. There are multiple points where concentrations are not obvious, or amounts added at steps are not given.

We thank the reviewer for pointing this out. All methods have now been updated for clarity and reproducibility. A new table including all the chemicals components with their catalogue and CAS numbers has been added to SI.

15) As is good practice, source data should be made available on a repository

A repository link with source data will be added.

REVIEWER COMMENTS

Reviewer #1 (Remarks to the Author):

I was pleased to re-review this manuscript by Yandrapalli et al. The authors have taken on board reviewer comments and strengthened their manuscript, particularly regarding the control experiments necessary to verify the new triple-layer inverted emulsion phase transfer method. Further experiments were also conducted that demonstrate control of vesicle domain curvature through binding of proteins to differing domains and faces of the vesicle. These additions have largely addressed my concerns, and I believe the manuscript is suitable for publication in Nature Communications after addressing the following:

- I could not see information on the lipid mass and concentration in mineral oil / squalene in the inverted emulsion method section – please add to improve reproducibility.

In the Methods section, the first sentence of the ‘Inverted emulsion method’ states that lipid concentration is 400 μM . To be clearer, we will reiterate it in the next sentence as below:

“The final lipid-oil mixture, with a concentration of 400 μM , is stored at 4 °C and used within a week.”

- Figure 2C lists GFP as inside the GUV and the NTA lipid on the outside. I believe the opposite is the case? Please correct.

You are correct. Thanks for pointing this out. We have replaced it with the correct illustration.

- Figure 3 – can you list the number of vesicles analysed for each chol% studied?

We have added this to the caption, now - $n \geq 70$

- Figure S4: Please show exemplar images for each of the compositions studied alongside the phase diagram.

The figure is now updated with the one below.

- Figure S6 is a very interesting addition to the manuscript. Could you please list the number of vesicles studied for each streptavidin concentration? This is missing.

That information ($n \geq 15$) is now added into the caption of the figure as well as in the main text.

- “In the Supplementary Figure S4(b), the Lo buds have a bright red ring (due to CT-B A1-594 binding) and are devoid of streptavidin (green) binding” – should be Supplementary Figure S7?

It is. It is now S8 after new additions. Thanks again for pointing it out.

- In the SI, in the “Shape parameters of two-domain vesicles” section, the final sentence ends stating there are five elastic shape parameters, but this sentence is not finished?

It is now corrected.

Using the vesicle size R_{ve} again as the basic length scale and the bending rigidity κ_o as the basic energy scale, we are then left with five elastic shape parameters: the dimensionless spontaneous curvatures $m_o R_{ve}$ and $m_d R_{ve}$ of the Lo and Ld domains; the ratio κ_d / κ_o of the two bending rigidities; as well the dimensionless Gaussian curvature difference and line tension $(\kappa_{Go} - \kappa_{Gd}) / \kappa_o$ and $\lambda R_{ve} / \kappa_o$.

- Figure S8: 10 vesicles are presented but the caption states 6 are shown.

Thanks for pointing it out, this is corrected now.

- Please check the naming of all equations in the manuscript and SI – I believe due to manuscript additions there are multiple equations mislabelled in the manuscript text.

Checked and updated where necessary. Thank you..

Reviewer #2 (Remarks to the Author):

The authors addressed all my concerns and questions. I think the new data and explanations improved the manuscript.

I have no more comments.

We thank the reviewer.

Reviewer #3 (Remarks to the Author):

I thank the authors for their extensive response and adding in the new data to the manuscript, although I still have a number of concerns and queries.

From my reading this manuscript mentions two main findings, that they produce higher degree of lipid bilayer asymmetry (1) and better phase separated asymmetric GUVs (2), compared to previous emulsion phase transfer methods.

For the first point, all reviewers asked for more comparison to previous methods, which the authors gave in the form of the double-layer method (Fig 1). Surprisingly, they only saw an improvement in the symmetry of the outside leaflet, which dramatically reduces their argument.

While the improvement in the degree of asymmetry for the inner leaflet is not statistically significant, the error bars are dramatically reduced which significantly improves the reproducibility of the method.

Furthermore, the authors did not even quantify the comparison of the triple layer (with an oil free layer) to triple layer (without an oil free layer). They did this with a visual inspection (Fig S1), which did show the oil free version kept the two layers separated somewhat, even after centrifugation. They need to actually measure the asymmetry to see if the oil free layer is required. Furthermore, in my original review I said the authors should also add an emulsion formed with the inner lipid+mineral oil (with time for the monolayer to form) to a phase transfer column prepared with the outer lipid+squalene (again with time for the monolayer to form). This again is likely to keep the asymmetry to a similar level and would be much simpler. The authors even state that “the rate of formation of lipid monolayer is much longer than the rate of droplet transfer”, therefore minimal lipid/oil mixing is likely not to cause a problem if the monolayers are already formed.

We appreciate the opportunity to clarify our methodology. This new method remains straightforward and is a rational extension of established emulsion phase transfer techniques. Our

method involves transferring an emulsion of [inner lipid + mineral oil] droplets through a column containing the [outer lipid + squalene] phase. The only modification is the use of squalene (for domains) and the inclusion of an intermediate, lipid-free mineral oil layer that rests on top of the squalene phase. This is not a complication but a critical simplification that serves two purposes. First to act as a physical barrier, that can still be seen after the centrifugation and movement of droplets through it. The second is to perform a washing step. As the droplets pass through this layer, it washes away any loosely associated lipids from the droplet surface before entering the squalene layer, further ensuring the purity and defined composition of the final outer leaflet. We cannot increase the incubation time for the water in mineral oil emulsion because they quickly begin sediment and fuse into larger droplets. This was fundamental issue with the previous method and why our inclusion of the spacer layer works to separate the unwanted free lipids from going to the other monolayer.

Furthermore, there is little quantification of protein binding in Fig 2 (just an individual GUV for each condition in Fig S2), could the other control methods provide similar results here even if their asymmetric values measured using NBD aren't as good (is protein being less sensitive than the NBD assay)?

In response to your valid criticism regarding the quantification in Figure 2, we have now updated the Supplementary Information (Figure S3) to include analysis of a larger population of GUVs for all conditions, moving beyond single representative images. This new data provides robust statistical support for our conclusions.

Figure S3: Confocal images of asymmetric GUVs with varying lipid compositions and protein interactions corresponding to the data shown in Figure 2 in the main text. Plots on the right showing the percentage of GUVs with protein binding for three distinct molecular recognition systems. For each system, binding was assessed in a positive control configuration (e.g., protein and its target lipid are accessible) and a negative control configuration (e.g., protein and its target lipid are inaccessible). Data is compiled from $N > 30$ vesicles for all conditions. Scale bars: 20 μm .

The following has been added into the main text:

Finally, we quantified the results for all three distinct molecular recognition systems in Figure 2. For each system, the positive control was when the protein in the external solution could access its target lipid on the outer leaflet, and the negative control was when the target lipid was situated on

the inaccessible inner leaflet (Figure S3). The results demonstrate a high degree of binding specificity. For the streptavidin-biotin system, we observed binding in 89.5% of the positive control GUVs, whereas only 6.6% of negative control vesicles showed a signal. The specificity was even more pronounced for the His-tag GFP-NTA system, which resulted in 100% binding for the positive control and 0% for the negative control. Finally, we tested the binding of the CT-B to its target ganglioside, GM1. A robust binding signal was detected in 97.0% of GUVs when GM1 was on the outer leaflet, compared to only 9.1% when it was on the inner leaflet.

For the second point, the improved phase separation seems to come entirely from the improved ability to add cholesterol into the membrane (from my reading). The authors state this is because of the use of squalene. However, as like the first point, the squalene is not reliant on the triple layer with an oil-free layer. Therefore, can similar results for this part also be obtained using triple layer without the oil-free layer or incubating the monolayers separately?

With respect to your second point, we have already tried squalene and other organic solvents as sole lipid mixtures. We published this information in our paper in 2019 (Moga et al. 2019). Squalene-based lipid mixture yields very few numbers of GUVs and at $>400 \times g$, only. In our experience, such high centrifugal speeds results in excessive vesicle clustering and bursting (Moga et al. 2019). Such low GUV numbers mean that the majority of droplets fail to convert into GUVs, resulting in excessive leakage of internal components into the outer solution and thus affecting the overall asymmetry that we are aiming for.

To make this clear, we have added the text below to the manuscript in the Discussion section:

However, employing squalene as the sole oil phase for the entire inverted emulsion process is not a viable strategy. As we have previously demonstrated, GUVs generated from a single squalene-oil phase exhibit very low formation efficiency and suffer significant leakage of their aqueous contents due to the required at high centrifugation speeds ($>400 \times g$) (Moga et al., 2019). This instability fundamentally undermines the goal of creating and maintaining a well-defined lipid asymmetry. To overcome these established limitations, we engineered the current oil system. Mineral oil is used for the initial droplet encapsulation, and as a physical barrier to prevent lipid mixing (also serving as a wash step to remove excess, non-incorporated dye). By reserving squalene for the destination phase, we decouple the requirements for stable emulsion formation from the final membrane composition. This design allows us to leverage the unique benefits of squalene for inducing phase separation (i.e. with cholesterol) without the associated penalties of poor yield and vesicle instability, thus enabling the reliable production of intact, asymmetric as well as phase-separated GUVs.

Overall, you can see that my concern is that the system is over-engineered, and not all of the parts are required. Or that similar results can actually be obtained in a simpler way than careful layering of four layers together.

We have carefully considered our options before compiling the oil layers, not over engineering. Our primary motivation is to achieve robust bilateral asymmetry - controlling the composition of both the inner and outer leaflets independently - not just one side. We validated this by demonstrating highly specific protein binding (also to domains) with three distinct protein-lipid pairs, which constitute an extensive functional evaluation of the system's specificity. Moreover, the choice of oil (i.e. squalene, which is denser than mineral) allows us to also achieve phase separation without additional oils keeping the system as simple as possible for both asymmetry and domains.

Other points:

For “Even with experimental observations lasting well beyond 6 h, the observed asymmetry and the resulting membrane curvature remained stable for the entire duration” the authors need to show this data.

To clarify, a typical experiment starts with centrifugation in a microtiter plate, followed by imaging under the microscope. On any given day, centrifugation is performed only once (by mid-day) and the vesicles in multiple wells will be imaged for the rest of the day (up to 6pm for example). So, most of the experiments are well beyond 1h and up to 6h.

The below example is shown for vesicles with Ni-NTA lipid on the inside at time 0h and again imaged after GFP addition at time 6h. If there is change in asymmetry, one should observe the binding of GFP to the Ni-NTA lipids, this is not the case even after 6h. A similar example for membrane curvature system is included and both have been added to the SI as a new Figure which is also referenced in the text. Moreover, videos S7 shows curvature as well.

Figure S11: Stability of asymmetric GUVs with and without asymmetry induced membrane curvature. Top panels, confocal cross-sectional images of vesicles containing Ni-NTA lipids on the inner leaflet at the start (left) and no binding of His-tag GFP is observed even after 6h (right). Bottom panels, membrane curvature of phase separated GUVs after CT-B binding at time 30 min (bottom, left) and sustained membrane curvature after 4h (right). Scale bars: 20 μm .

The supplier or preparation of His-GFP is missing from the text.

Thank you for noticing, it is now included in Table S1 of the SI.